

# Atmospheric turbulence observed during a fuel-bed-scale low intensity surface fire

**Joseph Seitz [1], Shiyuan Zhong [1*], Joseph J. Charney [2], Warren E. Heilman [2,*], Kenneth L. Clark [3], Xindi Bian [2], Nicholas S. Skowronski [4], Michael R. Gallagher [3], Matthew Patterson [4], Jason Cole [5], Mike T. Kiefer [1], Rory Hadden [6], and Eric Mueller [6]**

[1] Department of Geography, Environment and Spatial Sciences, Michigan State University, East Lansing, MI 48824; seitzjos@msu.edu (J.S.); mtkiefer@msu.edu (M.K.); zhongs@msu.edu (S.Z.)

[2] USDA Forest Service, Northern Research Station, 3101 Discovery Dr., Suite F, Lansing, MI 48910; joseph.j.charney@usda.gov (J.J.C.); warren.heilman@usda.gov (W.H.); xindi.bian@usda.gov (X.B.)

[3] USDA Forest Service, Northern Research Station, Silas Little Experimental Forest, 501 Four Mile Road, New Lisbon, NJ 08064; kenneth.clark@usda.gov (K.C.); michael.r.gallagher@usda.gov (M.G.)

[4] USDA Forest Service, Northern Research Station, 180 Canfield Street, Morgantown, WV 26505; nicholas.s.skowronski@usda.gov (N.S.); matthew.m.patterson@usda.gov (M.P.)

[5] USDA Forest Service, Northern Research Station, 5 Moon Library, 1 Forestry Dr., Syracuse NY 13210; jason.cole2@usda.gov (J.C.)

[6] The University of Edinburgh, Edinburgh, EH9, 3FB, UK; e.mueller@ed.ac.uk (E.M.); r.hadden@ed.ac.uk (R.H.)

\* Correspondence to: Shiyuan Zhong, zhongs@msu.edu; Tel.: +1-517-432-4743

Received: date; Accepted: date; Published: date





**Abstract.** The ambient atmospheric environment affects the growth and spread of wildland fires,
whereas heat and moisture release from the fires and the reduction of the surface drag in the
burned areas can significantly alter local atmospheric conditions. Observational studies on fire-
atmosphere interactions have used instrumented towers to collect data during prescribed fires,
but a few towers in an operational scale burn plot (usually $> 10^3$ m$^2$) have made it extremely
challenging to capture the myriad of factors controlling fire-atmosphere interactions, many of
which exhibit strong spatial variability. Here, we present analyses of atmospheric turbulence data
collected using a 4×4 array of fast-response sonic anemometers during a fire experiment on a 10
m × 10 m burn plot. In addition to confirming some of the previous findings on atmospheric
turbulence associated with low-intensity surface fires, our results revealed substantial
heterogeneity in turbulent intensity and heat and momentum fluxes just above the combustion
zone. Despite the small plot (100 m$^2$), fire-induced atmospheric turbulence exhibited strong
dependence on the downwind distance from the initial line fire and the relative position specific
to the fire front as the surface fire spread through the burn plot. This result highlights the
necessity for coupled atmosphere-fire behavior models to have 1-2 m grid spacing to resolve
heterogeneities in fire-atmosphere interactions that operate on spatiotemporal scales relevant to
atmospheric turbulence. The findings here have important implications for modeling smoke
dispersion, as atmospheric dispersion characteristics in the vicinity of a wildland fire are directly
affected by fire-induced turbulence.



## 1 Introduction

Wildland fires are directly affected by atmospheric conditions. Macroscale (thousands of
kilometers, weeks to months) atmospheric conditions such as prolonged periods without
substantial precipitation, high temperature, and low humidity that dry out and pre-heat fuels
often set background for large wildland fires (Potter, 1996; 2012; Finney *et al.*, 2015; Littell *et*
*al.*, 2016; Kitzberger *et al.*, 2017). Once ignited, fire behavior characteristics (e.g., burn intensity,
ember production, spotting, fire whirls and the rate of spread) are influenced more by microscale
(< 1000 m, < 1 h) conditions such as local topography and wind speed and direction at the
location of the fires. Most wildland fires tend to spread in the direction the wind blows, and the
stronger the wind speed the faster the fire spreads (Carrier *et al.*, 1991; Wolff *et al.,* 1991; Clark
*et al.,* 1996). Another essential microscale factor affecting fire behavior is atmospheric
turbulence, defined as irregular microscale air motions in the forms of eddies that are
superimposed on mean atmospheric motions (Stull, 1988).
Turbulent eddies affect fire behavior as well as the transfer of gaseous and particulate
emissions from the fires to the surrounding atmosphere (Clements *et al.*, 2008; Seto *et al.*, 2014;
Viegas and Neto, 2015; Skowonski and Hom, 2015; Heilman *et al.*, 2015; Heilman, 2021).
Turbulence in the atmosphere is generated primarily by wind shear as a result of changes in wind
speed and/or direction, known as mechanical turbulence, and by convection, referred to as
thermal turbulence. Mechanical turbulence is often generated when air flow encounters surface
drag, rough terrain or other natural or man-made obstacles and boundaries separating different
air masses (e.g., weather fronts), different land cover types (e.g., grass vs. forested land) or land
use types (e.g., agriculture vs. urban). Thermal turbulence is produced when heated surface air
rises up in the atmosphere, a process known as convection, which commonly occurs during



daytime when incoming solar radiation absorbed by the earth's surface exceeds outgoing
terrestrial radiation. Fire-induced turbulence is a type of thermal turbulence in that heat released
by combustion produces buoyant plumes that rise up from the combustion zone.
Despite the important role atmospheric turbulence plays in fire behavior and in the
exchanges of momentum and scalars (e.g., heat, moisture, carbon monoxide, carbon dioxide
particulate matter or PM) between the combustion zone and the surrounding atmosphere,
detailed observations of atmosphere turbulence in the presence of wildland fires have only
become available in recent decades. For instance, the first large-scale field experiment where
comprehensive turbulence data were collected above and in the vicinity of a wildland fire front
was FireFlux, conducted on February 23, 2006 over a 40-hectare plot of native tall-grass prairie
in Galveston, Texas (Clements *et al.,* 2007; Clements *et al.,* 2008). Fire-atmosphere interactions
were monitored primarily using fast-response three-dimensional (3D) sonic anemometers
mounted at multiple levels on a tall (43 m) and a short (10 m) tower within the burn plot. The
data revealed a fivefold increase in turbulence kinetic energy and a threefold increase in surface
stress during the fire-front passage, and a rapid return of turbulence to the ambient level behind
the fire front. A follow-up field experiment, known as FireFlux-II, took place at the same site in
2013, with more measurements designed to fill gaps in the original FireFlux experiment and
provide further information on fire–atmosphere interactions and fire-induced turbulence regimes
(Clements *et al.,* 2019). The data from FireFlux II have been used to validate fire behavior
models (Moody *et al.*, 2022), but the results on the intensive collection of turbulence data from
FireFlux II are yet to be reported in the peer-reviewed literature.
While the FireFlux and FireFlux II experiments in Texas provided direct turbulence
measurements during intense grass fires, a number of wildland fire experiments in the New



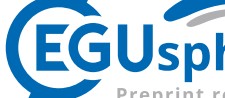

Jersey Pine Barrens provided information on fire-induced turbulence during low-intensity forest
understory fires (Heilman *et al.* 2015, 2017, 2019 and 2021; Mueller *et al.* 2017, 2019; Clark *et*
*al.* 2020). These experiments were conducted between 2010 and 2021 by research projects under
the auspices of the Joint Fire Science Program (http://www.firescience.gov) and the Department
of Defense Strategic Environmental Research and Development Program (SERDP)
(https://serdp-estcp.org/). The burn plots for these experiments, which were in the same areas of
the New Jersey Pine Barrens, ranged from about 5 to 100 hectares in size, with forest understory
vegetation (average about 1 m height) composed of blueberry, huckleberry and scrub oak and
overstory vegetation (average about 20 m height) composed of pitch pine and mixed oak.
Turbulence data were collected using 3D sonic anemometers and thermocouples mounted on a
20-m, a 10-m and a 3-m micrometeorological flux tower within the burn plots. The data from
these NJ fire experiments revealed large variations in turbulence intensity, stress, and fluxes
across the canopy layer, which complicated the evolution of local turbulence regimes and their
interaction with the spreading fires. Specifically, the data showed that fire-induced increases in
turbulent kinetic energy are considerably larger near the top of the forest canopy layer than
within the canopy, implying that vertical mixing or transport of fire emissions (e.g., PM,
moisture and heat) could be substantially larger near the canopy top than within the canopy layer
(Heilman *et al.,* 2015). The observations also revealed that an anisotropic turbulence regime
tends to persist throughout the vertical extent of overstory canopy layers, even within the highly
buoyant plume during the passage of fire fronts. The results suggested that spreading line fires
can have a substantial effect on the skewness of daytime velocity distributions typically found
inside forest vegetation layers, and that the contributions to turbulence production and evolution



from mechanical shear production and diffusion can be very different in the pre-fire and post-fire
environments (Heilman *et al.,* 2017).

The data from both the TX grass fires and NJ forest understory fires have also provided

insight into the turbulent momentum and heat transfer processes during the fires. The fire-
enhanced turbulence updrafts and downdrafts transfer warmer air (or lower momentum air) from
the surface upward, a process known as "ejection" and colder air (or higher momentum air)
downward to the surface, a process referred to as "sweep", which act to redistribute energy or
momentum between the combustion layer and the atmosphere above (Heilman *et al.*, 2021). The
analyses of the data from the TX and NJ fire experiments suggested that wildland fires in grass
or forest environments could substantially alter the relative importance of sweep and ejection
processes in redistributing momentum, heat and other scalars in the lower atmosphere (Heilman
*et al.,* 2021). For turbulent momentum transfer, sweep events were found to play a dominant role
at the fire front regardless of fire type despite the stronger updrafts than downdrafts at the front.
However, the effect of fires on turbulent heat transfer is different between the heading intense
grass fires and backing low-intensity forest-understory fires. The former tends to be dominated
by ejection events while in the latter case ejection and sweep events are equally important
(Heilman *et al.,* 2021).

Both the TX and NJ fire experiments mentioned above were conducted over plots on

relatively flat terrain. However, wildland fire behaviors can be affected significantly by
topography (Werth *et al.,* 2011; Sharples, 2009; Sharples *et al.,* 2012). This is because
topography exerts a strong influence on both weather and fuel conditions (Bennie *et al.*, 2008;
Ebel, 2013; Billmire *et al.*, 2014; Calviño-Cancela *et al.,* 2017; Povak *et al.*, 2018). A series of
prescribed burn experiments in California between 2008 and 2012 were conducted in complex



terrain with burn plots on a simple slope (Seto and Clements *et al.,* 2011; Seto *et al.*, 2013;
Clements and Seto, 2015; Amaya and Clements, 2020) or in a narrow valley (Seto and Clements,
2011). The burn plots in these experiments ranged from 2 to 15 hectares in size, but all plots
were dominated by grass fuels. Data from these experiments collected using
micrometeorological towers augmented by other remote sensing equipment provided unique
information on the interactions between terrain-induced circulations and fire-induced flows. The
results showed that terrain-induced slope flows and valley winds can interact with fire-induced
flows to enhance horizontal and vertical wind shears that subsequently contribute to turbulence
production. The interactions of fire-induced flows with slope winds also produce local
convergence or divergence with strong updrafts and downdrafts. Turbulence regimes tend to be
anisotropic immediately above fire fronts, moving towards isotropic conditions higher up (Seto
*et al.,* 2013, Clements and Seto, 2015; Amaya and Clements, 2020). The data from these studies
also revealed an increase in turbulent energy in both velocity and temperature spectra at higher
frequencies, as fire fronts shed small eddies, and an increase at lower frequencies that are related
to the strengths of the cross-stream wind component generated by the fire and enhanced by
topography (Seto *et al.*, 2013).

The aforementioned field experiments were conducted on operational-scale (or

management-scale) burn plots that ranged from several to 100 hectares and it was not feasible to
cover such large burn plots with just a few micrometeorological towers. Consequently, the
measurement strategy of these experiments was centered around tall towers placed at couple of
key spots in the burn plot to provide information on vertical variations of fire-atmosphere
interactions. The lack of spatial coverage of the complex fuel and atmospheric conditions at
these large burn sites makes interpretation of the limited observations challenging. Laboratory



studies (e.g, Forthofer and Goodrick, 2011; Campbell-Lochrine *et al.*, 2021; Di Cristina *et al.*,
2022) have the advantage of monitoring the fires using densely spaced instruments. However,
laboratory studies are often conducted under controlled conditions not necessarily representative
of the real fuel and atmospheric environments encountered in outdoor wildland fires. There
exists an apparent gap in the observations of fire-atmosphere interactions between operational-
scale burns and fine-scale laboratory experiments.

Here, we present analyses of turbulent data collected during a small-scale (10 m ×10 m)

experimental burn in the field that was densely instrumented for the purpose of bridging the gap
in our knowledge about fire-atmosphere interactions between operational-scale ($\geq 10^3$ m$^2$) and
laboratory-scale ($< 10^1$ m) fire experiments. The primary question we aim to address is how a
low-intensity surface fire may modify turbulence in the atmosphere just above the combustion
zone. More specifically, our analyses will explore the following questions: How does the surface
fire alter turbulence intensity and turbulent heat and momentum exchanges between the
combustion zone and the atmosphere above? Whether and how would the fire change the
partitioning of the heat and momentum fluxes into different types of events (both event number
and event contribution)? How does the modifications of the fire on turbulence vary spatially
across the burn plot? Answers to these questions could prove useful for predicting fire-
atmosphere interactions, particularly the momentum and scalar exchanges between the fire and
the atmosphere. Moreover, the answer to the last question could provide guidance regarding
what horizontal grid spacing in coupled atmosphere-fire behavior models is necessary to capture
horizontal variability in near-surface atmospheric turbulence during the presence of surface fires.




## 2 Method

### 2.1 Experiment and Instrumentation

The experimental burn that this study focuses on took place on May 20, 2019 in a pitch

and loblolly pine plantation at the Silas Little Experimental Forest in New Lisbon, New Jersey.

This particularly burn was part of broader series of 35, densely instrumented, low-intensity

surface fire experiments on 100 $m^2$ (10 m x 10 m) plots in this plantation conducted between

March 2018 and June 2019 by a SERDP research project that set out to collect data using

laboratory-scale ($10^0$-$10^1 m^2$) experiments, intermediate or fuel-bed-scale ($10^2$ $m^2$) burns and

management-scale ($10^{3-4}$ $m^2$) prescribed fires to improve the understanding of combustion

processes and fire-atmosphere interactions across scales (Gallagher *et al.*, 2022; Skwonski, *et al.*,

2021).

As shown in Figure 1, the 100 $m^2$ burn plot was densely monitored by instruments

mounted on four parallel east-west-oriented trusses (A, B, C, D). On each truss, four 3D fast-

response sonic anemometers (R.M. Young 81000V, Traverse City, MI, USA) were mounted at

2.5 m above the ground level (AGL) to collect the east-west ($u$), north-south ($v$) and vertical ($w$)

velocity components and temperature at a sampling rate of 10 Hz (Clark *et al.*, 2022a).

Additional 10-Hz temperature data were also obtained using fine-wire thermocouples (Omega

SSRTC-GG-K-36, Omega Engineering, Inc., Stamford, CT, USA) mounted at a range of heights

(0, 5, 10, 20, 30, 50, 100 cm) below the two inner trusses (B and C) (Clark *et al.*, 2022b). A

radiometer/visible spectrum camera pair was mounted adjacent to each sonic anemometer to

measure radiative heat fluxes and flame arrival times and persistence (Kremens *et al.*, 2022).

Spatially explicit fire spread data were derived from infrared data collected by an infrared video-

camera (A655SC, FOL6 100.0-650.0 C lens, FLIR Systems Inc., Wilsonville, OR, USA)



mounted on top of a 10-m tower in the center of the plot (Skowronski *et al.*, 2022a). A custom
field calorimetry hood (labeled TACO next to B2) with an inlet oriented over a portion of the
fuel bed was used to sample $O_2$, $CO_2$, and CO concentrations in buoyant plumes (Campbell-
Lochrie *et al.*, 2022). Gas concentrations were measured at 1 Hz using an Infrared gas analyzer
(Crestline NDIR 7911, Crestline, Livermore, CA, USA).

The analyses here focused only on the data from the 4×4 sonic anemometer array. All

sonic anemometer data underwent a quality assurance and control process to remove spurious
values (Clark *et al.*, 2022a). Initially, data that were collected prior to a designated common start
time was removed, providing a starting point for the observations for the burn period. Next, the
data from sonic anemometers include a self-reporting diagnostic column where any non-zero
number is considered an invalid measurement, so any measurement that reported a non-zero
diagnostic code was removed. Following these initial steps, data that fell outside the sonic
anemometer operating parameters (wind speed: ±40 m/s; temperature: ± 50 °C) were also
removed.

The horizontal wind velocities were rotated into a streamwise coordinate system where

the *u*-component (streamwise component) is aligned with the prevailing wind direction, and the
*v*-component (cross-stream component) is perpendicular to the prevailing wind direction pointing
to the left. Vertical winds were not corrected for tilt because of the short (<30 min) observational
period and because the burn plot was on level ground and each sonic anemometer was carefully
mounted and leveled so that the wind sensors were very close to true horizontal and vertical
planes. The results (presented below) indeed suggested that the contamination of vertical
velocity by horizontal velocities were negligibly small as the average vertical wind component
during the pre-burn period was nearly zero.




## 2.2 Fuel and ambient atmospheric conditions

The primary fuel for this burn was pitch pine needles (*Pinus rigida* Mill.). Based on

biometric and terrestrial laser scan measurements collected pre- and post-burn, the fuel mass was
estimated to be about 0.5 kg m$^{-2}$ and fuel moisture content about 5.5% (Skowronski *et al.*,
2022b).

The ambient atmospheric conditions on the day of the burn is indicated using the data

from a surface weather station located approximately 200 m northeast of the burn plot that has
similar type of land cover as the burn plot (Figure 2). Ambient winds were very weak in the
morning, varying in direction between south and west. Wind speeds increased in midday to about
5 m s$^{-1}$ along with a direction shift to southwest and west. This wind speed increase was likely
due to the mixing of higher winds from above to the surface as the mixing layer grew higher
during the day. The growth of the mixing layer was a result of increased turbulent mixing
associated with surface heating, as indicated by an increase in surface temperatures from about
20 °C in the morning to slightly above 30 °C around 1400 Local Standard Time (LST) and a
corresponding decrease in relative humidity from over 80% in the morning to less than 40% in
the early afternoon.

## 2.3 Fire spread

The experiment started around 14:25 LST when a single 10-meter cotton cord was

soaked in accelerant, ignited and then dropped on the fuel bed to produce a single, near linear
ignition across the western border of the plot. Infrared imagery data (Figure 3) captured by the
overhead infrared camera is used to evaluate the changes in temperature from just before ignition



(Figure 3a), immediately after ignition (Figure 3b), and through the period following the ignition
as the line fire spread with winds across the plot (Figure 3c-f). The average fire spread rate
throughout the burn was evaluated from these data to be approximately 5.4 cm s$^{-1}$. The ignition
produced a line fire parallel to the western boundary of the plot (Figure 3b). The line fire spread
in the direction of the west-southwesterly background wind towards the east-northeast over the
next few minutes (Figure 3c, d). The initial spread was faster on the northern portion of the
domain, as expected from the south-southwesterly wind direction. As the fire burned through the
northern portion of the plot, the fire front caught up in the southern portion (Figure 3e). The fire
ended at around 14:32:16 LST as the fire front reached the eastern boundary of the plot and ran
out of fuel to continue (Figure 3f).

**2.4 Data Analysis**

The quality-controlled 10-Hz wind and temperature data from the 3D sonic anemometers are

used to calculate turbulent perturbations defined as the differences between the instantaneous
observations and the mean values:
$$\varphi' = \varphi - \overline{\varphi} \qquad (1)$$
where $\overline{\varphi}$ is the mean value that is estimated by block-averages
$$\overline{\varphi} = \sum_{n=1}^{N} \varphi_n \qquad (2)$$

Here, N is the number of samples over the averaging period or the time block and the mean
values represent the mean state of the atmospheric flow. In traditional turbulence studies, mean
state is usually determined by averaging the data over a period of a few minutes up to 1 hour,
depending on atmospheric stability and the scale of interest. However, the block-averaged values
during the period of active burning are likely to be contaminated by the fire and therefore poorly



represent the mean background flow. To resolve this issue, Seto *et al.* (2013) and Heilman *et al.*
(2015) proposed that the block-averaged means for the fire period be replaced by block-averaged
means calculated during the pre-burn period. In order to adopt this approach, the observational
period is divided into three periods representing pre-burn, burn and post-burn, which are
described in detail below.

The arrival of the fire front at most locations in the sonic anemometer array was clearly

marked by a sharp rise in temperature (Figure 4). However, the magnitudes of the temperature
increase and the rates of increase vary with the location of the sonic anemometers because the
shape of the flame front was irregular (Figure 3). Note that the sonic temperatures are limited to
50 °C, which is the operational range for the instruments beyond which data are deemed
unreliable. Based on the temperature time series and the time when the fire was ignited along the
western boundary (14:25 LST), the 10-min period from 14:15:13 through 14:25:12 LST is
defined as the pre-burn period over which the mean values for $u$, $v$, $S$ (horizon wind speed), $w$,
and $T$ are calculated, and these values are used for computing perturbations for the entire
experiment. The definition of the burn period, however, is complicated by the fact that the fire
front reaches/leaves each sonic anemometer at a different time and consequently the true burn
period across the plot varies somewhat depending on the location of each sonic anemometer.

To create a robust definition of the burn period that can be applied to all the sonic

anemometers in the 4 × 4 array, and eventually to other burns in the broader burn series, the
sharp rise in sonic temperatures associated with fire front is measured using integer ($n$) multiples
of the standard deviation (denoted using $\sigma$) of the average temperature over the pre-burn period.
A threshold value that is too small (e.g., 1 or 2 times standard deviation) may not distinguish the
increase in temperature associated with the fire front from normal temperature fluctuations





during the day, but a value that is too large (e.g., 10 time standard deviation) may fail to detect
the fire front associated with a small or moderate temperature increase. Figure 5 shows the
number of sonic anemometers whose temperatures exceed $n \times \sigma$ as $n$ increases from 1 to 35, and
the length of the exceedance period. As $n$ increases from 1 to 8 or the threshold value for fire-
induced temperature increase changes from $1\sigma$ to $8\sigma$, the number of sonic anemometers drops
from 16 to 13 and the period drops sharply from just under 60 min to about 6 min. Continued
increases in the threshold values from $8\sigma$ to $25\sigma$ result in no change in the number of
anemometers and very little change in the length of the period (less than 1 min). This analysis
suggests that $8\sigma$ can be used as the threshold for temperature increases associated with fire front.
Thresholds lower than $8\sigma$ would imply a burn period of 30- to 60-min long that, according to the
time series in Figure 4, would include periods of no fire and therefore de-emphasize the effects
of the fire in the resulting analyses. Applying this criterion to all the sonic anemometers and
defining the burn period as between the first and last sonic temperature at or above the threshold
leads to the selection of the burn period as 14:26:13 to 14:32:29 LST. Finally, the 10 min
following the burn period (14:32:30 to 14:42:29 LST) is defined as the post-burn period.

Following the establishment of the three periods, wind and temperature perturbations are

calculated using equations (1) and (2), where the pre-burn averaged values are used as means for
the burn and post-burn periods. Strictly speaking, the perturbations calculated for the burn and
post-burn periods are not classical turbulent perturbations; to differentiate the features from
classical turbulence, they should be interpreted as being primarily fire-induced turbulent
perturbations.

As noted above, horizontal wind velocity is rotated into a streamwise coordinate where

the $x$-component (streamwise component, $u$) is aligned with the prevailing wind direction and the





*y*-component (cross-stream component, *v*) is perpendicular and pointing to the left of the
prevailing wind. The prevailing wind direction for the rotation is determined by the 10-min pre-
burn period average of wind directions across all 16 sonic anemometers. The average wind
directions during the pre-burn period vary slightly across the 16 sonic anemometers, with mean
and median wind directions of 225 and 226 degrees, respectively. The subtle variations in wind
directions is possibly due to slight error in sensor alignment, rather than actual flow
heterogeneity. The 226 degrees is used as the prevailing wind direction for the purpose of
coordinate rotation.

The quality controlled, coordinate rotated data from the sonic anemometers are analyzed

to determine fire-induced changes to turbulence intensity, vertical heat fluxes and vertical fluxes
of horizontal momentum also known as shear stress just above the combustion zone by
comparing values between the pre-burn and the burn periods. The values are also compared
between the pre-burn and post-burn periods to determine how quickly the effects of fire dissipate
or how fast the atmosphere returns to the ambient state.

Turbulence intensity is measured by the turbulent kinetic energy (*TKE*) defined as the

sum of the variance of the three velocity components:

$$TKE = \left( \overline{u'^2} + \overline{v'^2} + \overline{w'^2} \right)/2 \qquad (3)$$


Turbulent shear stress is commonly measured by shear velocity or friction velocity denoted by
$u_*$ and the square of friction velocity is related to the magnitude of the kinematic vertical flux of
horizontal momentum:
$$u_*^2 = \left( \overline{u'w'}^2 + \overline{v'w'}^2 \right)^{\frac{1}{2}} \qquad (4)$$




where u'w' and v'w' are the vertical fluxes of streamwise and corss-stream momentum flux,
respectively and the overbar denotes time average. The average period is 1 min for this analysis
to be consistent with previous studies on fire-induced turbulence (Seto *et al.*, 2013; Heilman *et*
*al.* 2021). Vertical heat flux is calculated as $\overline{T'w'}$ and the averaging period is also 1 min.

For the analyses of vertical turbulent fluxes of heat and horizontal momentum, a quadrant

analysis technique (Katul *et al.,* 1997, 2006; Heilman *et al.,* 2021) is utilized to delineate the
contributions to the turbulent heat or momentum transfer from four types of processes
corresponding to the four quadrants of a *w'*(horizontal) and $\varphi'$ (vertical) coordinate, where the *w'*
denotes vertical velocity perturbation and $\varphi'$ denotes perturbations of temperature (*T'*) or
horizontal wind speed (*S'*) in heat or momentum flux calculations, respectively. The four
quadrants are: Q1: $\varphi'w' > 0$, $\varphi' > 0$, $w' > 0$; Q2: $\varphi'w' < 0$, $\varphi' > 0$, $w' < 0$; Q3: $\varphi'w' > 0$, $\varphi' <$
$0$, $w' < 0$; Q4: $\varphi'w' < 0$, $\varphi' < 0$, $w' > 0$. Note that the perturbation in horizontal wind speed
(*S'*), rather than the streamwise or cross-wind components (*u'* or *v'*), are used for computing
momentum flux following Heilman *et al.*, (2021):

$$S' = S - \overline{S} \qquad (5)$$
$$S = \sqrt{u^2 + v^2} \qquad (6)$$

The quadrant analysis is also known as sweep-ejection analysis (Heilman *et al.*, 2021)

which associates each quadrant with a specific type of vertical turbulent transfer events. The
names of the events and the associated quadrant designations, which are different for turbulent
heat and momentum fluxes, are given in Table 1.

Based on the definition in Table1, ejection (Q1) and sweep (Q3) events contribute to

positive vertical turbulent heat flux through the upward transfer of warmer air from below



(ejection) or the downward transfer of cooler air from above (sweep), while inward interaction
(Q2) and outward interaction (Q4) events contribute to negative turbulent heat flux through the
downward transfer of warmer air from above (inward interaction) or the upward transfer of
cooler air from below (outward interaction). For vertical flux of horizontal momentum, inward
interaction and outward interaction events contribute to positive flux through the upward transfer
of faster moving air (outward interaction) or the downward transfer of slower moving air (inward
interaction), while sweep and ejection events contribute to negative momentum flux through the
downward transfer of faster moving air (sweep) or the upward transfer of slower moving air
(ejection). Note that the warmer/cooler or faster/slower air is relative to the air in the adjacent
layers.

The sweep-ejection analysis calculates the proportion of a given type of events by simply

counting the number of events or the data points in the 10 Hz time series that fall within the
given quadrant. The contributions of the given type of events to the average turbulent fluxes over
a given time period ($T_p$) are calculated, following Heilman *et al.* (2021), by the integral

$$\overline{\varphi'w'}_Q = \frac{1}{T_p}\int_0^{T_p}\varphi'(\tau)w'(\tau)\varepsilon_Q d\tau \qquad\qquad (7)$$

where $\varepsilon_Q$ is 1 for the given quadrant and zero otherwise, $\tau$ is time and $\varphi'$ is temperature or
horizontal wind speed perturbation for heat or momentum fluxes, respectively.

**3 Results and Discussion**

**3.1 Fire-Induced Perturbations to Wind and Temperature**



Before we examine fire-induced changes to turbulence in ambient atmosphere, we first
take a look at the response of the instantaneous temperature and wind to the surface line fire
recorded by the 16 sonic anemometers as the fire spread from west to east across the 10 m ×10 m
burn plot (Figure 6). Note that perturbation temperatures ($T'$, see Eq. 1), instead of actual
temperatures, are shown to accommodate the magnitude difference between temperature and
wind and therefore making it easier to visualize jointly the effects of the fire on temperature and
wind. The natural or non-fire fluctuation recorded during the pre-burn period are small, with
magnitudes generally less than 2.5 m s$^{-1}$ for $u$, 1 m s$^{-1}$ for $v$ and 2.5 $^{\circ}$C for $T'$. The fire impinging
upon the sonic anemometers is marked by a sharp increase in $T'$, but the magnitude of the
temperature changes depend heavily on location, from very little change on the western side (A1,
B1, C1, D1) of the burn plot where the fire was ignited, to a nearly 20$^{\circ}$C increase on the eastern
side (A4, B4, C4, D4). This spatial heterogeneity in $T'$ is consistent with the pattern of the fire
spread from the western boundary toward the east and northeast by the southwesterly ambient
wind (Figure 4). During the burn period, the $u$ fluctuations decreased slightly while the $v$
fluctuations increased. The $v$-component no longer fluctuated around zero, as in the pre-burn
period, but rather it was dominated by negative values, indicating a systematic shift in wind
direction. There was a tendency for $u$ and $T'$ to return towards the pre-burn conditions after the
burn, but the $v$ component remained negative during the post-burn period.
The observed changes in the distribution of wind and temperature values associated with
the fire at all 16 sonics are summarized by the box-whisker plots in Figure 7. The pre-burn mean
is 1.7 m s$^{-1}$ for the streamwise wind component $u$ and near zero (-0.04 m s$^{-1}$) for the cross-stream
component $v$. The pre-burn vertical velocity distribution also has near zero mean, which
confirms that the sonic anemometers were well-leveled. During the burn period, the mean of $u$



dropped in magnitude from 1.7 to 1.05 m s$^{-1}$ while the mean of $v$ increased in magnitude from -
0.04 to -0.65 m s$^{-1}$, indicating an overall shift in wind direction from southwesterly to west-
southwesterly. This change in the horizontal wind components suggests that ambient air was
drawn towards the fire producing convergence at the fire front. There is also a fire-induced
widening of the distributions of the horizontal wind components, particularly the $v$ component,
and an increase in the number of outliers with magnitudes that nearly doubled the pre-fire
magnitude. The large negative values in $v$ during the burn period reinforce the suggestion of
convergence in the vicinity of the fire.

Interestingly, there is little evident change in the overall distribution of $w$ during the burn

period, except that more and larger outliers are indicated. The maximum updrafts (downdrafts)
during the burn period reach speeds of nearly 6 m s$^{-1}$ (-5 m s$^{-1}$), which is more than double those
of the pre- and post-burn periods, suggesting that intermittent turbulent eddies associated with
the fire could have a strong impact on vertical velocity just above fuel bed. The $T$' distribution
also widens substantially during the burn period ($\sigma$=4.24 ℃) compared to the pre-burn period
($\sigma$=0.48 ℃), with the maximum temperature perturbation reaching nearly 20°C.

The influence of the fire on the horizontal wind components continues into the post-burn

period, as the post-burn distributions of $u$ and $v$ fall between those of the pre-burn and burn
periods. In contrast, the post-burn $w$ distribution returns to a distribution very close to that of the
pre-burn period. Similarly, the $T$' distribution during the post-burn period is very similar to that
of the pre-burn period. The similarities between the $w$' and $T$' distributions suggest that the two
variables are closely related to each other, with large updrafts during the burn period being
generated primarily by heating. This result suggests that the fire-induced circulation exhibits



behavior more consistent with a buoyant plume than mechanically forced rising motion resulting
from converging surface air.

### 3.2 Intensity of Fire-Induced Turbulence

We now explore the modifications of the fire to atmospheric turbulence properties just
above the combustion zone. The first question to address is how turbulence intensity quantified
by *TKE* in Eq. (3) is modified by the fire and how the modification may vary with location in the
burn plot. Figure 8 shows time series of 1-minute averaged *TKE* and its three components (the
variance of the three velocity components) for each of the sonic anemometers. The time series
indicate lower *TKE* values in the pre-burn period, larger values during the burn period, and
values remaining high in the post-burn period. The burn period *TKE* is primarily driven by an
increase in horizontal velocity variance, $\overline{u'^2}$ and $\overline{v'^2}$, particularly the cross-stream component
$\overline{v'^2}$. The *TKE* values remain high into the post-burn period and, at several sonic anemometers
(D3 and C4), the post-burn *TKE* peaks are comparable with or higher than the peaks observed
during the burn period.
The box-whisker plots in Figure 9 depict the fire-induced changes to the distribution of
turbulence intensity as observed by all 16 sonic anemometers. Averaging across all the
instruments, the burn period mean *TKE* is 1.25 $m^2s^{-2}$, which is roughly double the pre-burn mean
of 0.697 $m^2s^{-2}$. The interquartile range of the burn period *TKE* is nearly three times the pre-burn
period range. Despite the increase in the mean and the interquartile range of the *TKE* from the
pre-burn to the burn period, the mean *TKE* values are still below 3 $m^2s^{-2}$, which is a threshold
sometimes used as an indicator for substantial boundary-layer turbulence (Stull, 1988; Heilman
and Bian, 2013), suggesting that this low-intensity surface line fire fails to produce a



substantially turbulent environment at the levels just above the fuel bed. The mean *TKE* in the
post-burn period does not return to that of the pre-burn period and remains elevated (1.21 m$^2$s$^{-2}$).
While the $\overline{w'^2}$ returns to the pre-burn conditions, the horizontal components remain elevated.

More specifically, $\overline{u'^2}$ and $\overline{v'^2}$ make up 53.0% and 38.5% of the average pre-burn *TKE*,

respectively. During the burn period, the contribution to *TKE* from $\overline{u'^2}$ decreases slightly to
49.1% and the contribution from $\overline{v'^2}$ increases substantially to 43.3%. As noted earlier (Figures 6
and 7), the burn period also exhibits a larger range of horizontal and vertical wind components,
which is consistent with the larger range of *TKE* values in Figure 9.

In the post-burn period, the distribution of vertical velocity variance returns to the pre-

burn distribution. However, the range of values in the horizontal components are smaller during
the post-burn period than the burn period, but still larger than during the pre-burn period. The
medians of the horizontal *TKE* components are higher in the post-burn period than in either of
the other periods. While the $\overline{u'^2}$ outliers (above the 99.3rd percentile) decrease, the $\overline{v'^2}$ outliers
increase in magnitude. As was previously discussed, post-burn average wind directions differ
slightly from the pre-burn, accompanied by increases in the magnitude of the horizontal winds
(Figures 6 and 7). This result is consistent with elevated *TKE* values persisting into the period
after the end of the fire.

Additional analysis of the variance of the three velocity components enables an

assessment of turbulence anisotropy indicated by the ratio of $\overline{w'^2}$ to 2x*TKE*. When this ratio
approaches 1/3 for a given time period, the period can be said to experience an isotropic
turbulent regime (Heilman *et al.*, 2015). The mean $\overline{w'^2}$ for all the sonic anemometers is 0.0597
m$^2$s$^{-2}$ for the pre-burn period, 0.0931 m$^2$s$^{-2}$ for the burn period, and 0.052 m$^2$s$^{-2}$ for the post-burn





period, which yields an anisotropy ratio of 0.042, 0.036, 0.021 for the pre-burn, burn and post-
burn periods, respectively. As the anisotropy ratios are well below 1/3 in all three periods, the
turbulence regime just above the combustion zone remains anisotropic at all time. It is worth
noting that in contrary to the belief that the increase in vertical velocity variance in response to
the surface heating during the burn should act to move turbulence towards a more isotropic
regime, the ratio here is slightly smaller during the burn period than the pre burn period largely
because the fire-induced increase in the cross-stream velocity variance is larger than the increase
in the vertical velocity variance. Heilman and Bian (2015) calculated the anisotrophy ratios at 3
m above ground for two forest understory fires. The ratio decreased from 0.118 to 0.0718 from
pre-burn to burn in one experiment, but increased from 0.089 pre-burn to 0.13 in another
experiment. Since the sonic anemometers located on the western and southern sides of the burn
plot show no clear increase in $\overline{w'^2}$, the anisotropy ratio is also calculated for each sonic to verify
that the mean values did not mask anisotropy variations at individual locations in the burn plot.
No individual sonic anemometer reaches a ratio of 1/3, and the highest individual ratio (0.133) is
found at sonic anemometer A4 during the burn period. This result indicates that overall, the *TKE*
just above the combustion zone is highly anisotropic and is dominated by the horizontal
components for this burn. This result is not surprising as the sonic anemometers are located only
2.5 m above ground where horizontal turbulence would be expected to dominate over vertical
turbulence (Heilman *et al.*, 2015).

**3.3 Fire-Induced Shear Stress**

To address the question on how the surface fire alter turbulent momentum transfer

between the combustion zone and the atmosphere above, we next explore fire-induced changes
to turbulent momentum fluxes or shear stress measured by friction velocity described in Eq. (4).





Figure 10 shows time series of 1-minute averaged $u_*^2$ and the streamwise $\overline{u'w'}$ and cross-
stream $\overline{v'w'}$ stress components (the momentum flux), measured by each of the sonic
anemometers for the three periods. Kinematic momentum fluxes and $u_*^2$ are similar across all
the sonic anemometers during the pre-burn period, although three of the northernmost
instruments (A2, A3, and A4) indicate a negative spike in $\overline{u'w'}$ just before the start of the burn
period. These spikes contribute to an increase in $u_*^2$ at this time as well. It is not clear what
caused these features, but candidates include an anomalous burst of wind along the northern edge
of the burn plot and possible contamination of the wind data by activities of the burn managers
as they prepared to ignite the fire.
During the burn period, the values of $\overline{u'w'}$ and $\overline{v'w'}$ increase somewhat, leading to
increases in the $u_*^2$ values. The fire-induced changes generally increase in magnitude from west
(left) to east (right) and south to north, consistent with the fire-spread pattern. The largest
increase occur at the easternmost (right) locations, particularly A4 and C4 where $u_*^2$ values
nearly doubled. The smallest increases are not found at the westernmost locations, but at C2 and
D2. With a few exceptions, $\overline{u'w'}$ and $\overline{v'w'}$ are negative in the beginning of the burn period,
turning positive later in the period. The $\overline{u'w'}$ values exhibit the largest burn period variation at
A4, followed by B4, and similar patterns are observed for $\overline{v'w'}$. Overall, variations in $u_*^2$ suggest
an increase in shear stress magnitude in the burn period compared to the pre-burn period, with
the easternmost sonic anemometers recording 1-minute averaged values that are far greater than
the westernmost sonic anemometers.
During the post-burn period, some sonic anemometers (A2, B2, C1, C2, D2) recorded
higher $u_*^2$ than during the burn period, while others (A1, B1, B3, C2, C3, D3) recorded values



similar to the burn period. In either case, the average values are larger than during the pre-burn
period. The maximum post-burn values among all the sonic anemometers occur at A2 for $u_*{}^2$
and $\overline{v'w'}$ and C1 for $\overline{u'w'}$, both of which are larger than their burn-period peaks.

The overall distributions of $u_*{}^2$, $\overline{u'w'}$, and $\overline{v'w'}$ from all 16 sonic anemometers are

depicted in Figure 11. During the pre-burn period, $\overline{u'w'}$ is negative, with a mean value of -0.015
$m^2\ s^{-2}$, indicating an overall downward transfer of higher streamwise momentum air, which is
expected as wind speed usually increases with height. The mean of the cross-stream momentum
flux $\overline{v'w'}$ is near zero (0.007 $m^2\ s^{-2}$). However, the spread of the two components is similar, with
standard deviations of 0.057 $m^2\ s^{-2}$ and 0.046 $m^2\ s^{-2}$ for $\overline{u'w'}$ and $\overline{v'w'}$, respectively. The pre-burn
stress $u_*{}^2$ of 0.061 $m^2\ s^{-2}$ ($u_* = 0.25\ m^2\ s^{-2}$) is typical for daytime surface layers.

An increased in the downward (upward) transfer of higher streamwise (cross-stream)

momentum is observed during the burn period as the median values become more negative for
$\overline{u'w'}$ and more positive for $\overline{v'w'}$. However, the mean values change little from the pre-burn
period. The spread is doubled from a standard deviation of 0.046 to 0.098 $m^2\ s^{-2}$ for $\overline{u'w'}$ and
nearly tripled from 0.05 to 0.124 $m^2\ s^{-2}$ for $\overline{v'w'}$. The stronger upward transfer of cross-stream
momentum is consistent with the generation of cross-stream wind and updrafts in the vicinity of
the surface fire. Despite this overall fire-induced increase in $\overline{v'w'}$, the distribution of the cross-
stream momentum is negatively skewed by large negative outliers, suggesting occasional transfer
of higher cross-stream momentum by downdrafts near the vicinity of the fire. Both the mean and
standard deviation of $u_*{}^2$ values are doubled to 0.13 $m^2s^{-2}$ and 0.086 $m^2s^{-2}$, respectively, over the
pre-burn values. The peak 1-min averaged values of $u_*{}^2$ exceed 0.4 $m^2s^{-2}$ (or a friction velocity
of 0.6 $m\,s^{-1}$), which is 2.5 times larger than the pre-burn values. Clements *et al.* (2008) also

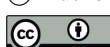



observed a three-fold increase in friction velocity in their experiment involving a high intensity
grass fire, although the absolute values of the friction velocity in their experiment were five
times larger (1 and 3  m s$^{-1}$ before and during the fire) than the current experiment.

The mean post-burn $u_*{}^2$ value (0.10 m$^2$s$^{-2}$) is lower than that of the burn period but still

higher than the pre-burn value, driven primarily by the cross-stream component. The values of
the $\overline{v'w'}$ (0.0471 m$^2$ s$^{-2}$) in the post-burn period is more than six times the pre-burn average
(0.0072 m$^2$s$^{-2}$), with a standard deviation (0.069 m$^2$s$^{-2}$) that is between the pre-burn period
(0.046) and burn period (0.096) values. The mean friction velocity therefore does not return to
the pre-burn average, although it is lower than the average during the burn period. Other
experiments (e.g. Clements *et al*, 2008; Heilman, *et al.* 2019) noted a return of friction velocity
to pre-burn values soon after the passage of the fire front, during a period when smoldering was
occurring. The results of this analysis suggest that friction velocities do not quickly return to pre-
burn values on all fires.

**3.4 Fire-Induced Turbulent Heat Flux**

We proceed to examine the impact of the fire on turbulent heat flux. Time series of 1-

minute average turbulence sensible heat flux $\overline{T'w'}$ for each sonic anemometer are shown in
Figure 12 for the three periods, which also shows the overall distribution of heat fluxes for all the
sonic anemometers. In the pre-burn period, the sonic anemometers recorded background $\overline{T'w'}$
values that averaged around 5.25×10$^{-2}$ ºC m s$^{-1}$ (or 52.7 W m$^{-2}$after multiplying by the density
and heat capacity of air), with a standard deviation of 3.41×10$^{-2}$ ºC m s$^{-1}$ (34 W m$^{-2}$). During the
burn period, a fire-induced increase in $\overline{T'w'}$ is evident at all but the westernmost sonic





anemometers (A1, B1, C1, and D1), with larger increases appearing at the easternmost locations.
The largest $\overline{T'w'}$ values generally occur early in the burn period, with the A4 sonic having the
largest $\overline{T'w'}$ value of 2.13 °C m s$^{-1}$ (2.138 kW m$^{-2}$). Based on the IR imaging (Figure 4), after the
first three minutes of the burn period there is a slight shift in the burn direction towards the
southeastern side of the plot. This shift in direction is apparent in the time series for the D4 sonic
anemometer, which is located on the southeastern corner of the burn plot, where elevated $\overline{T'w'}$
values are recorded late in the burn period, at a time when the values have dropped at most of the
other sonic anemometers. The overall distribution of the burn-period $\overline{T'w'}$ is skewed by larger
values since the plot mean was 0.268 K m s$^{-1}$ (269 W m$^{-2}$) but the median was just 0.0974 °C m
s$^{-1}$(98 W m$^{-2}$).
Values of $\overline{T'w'}$ during the post-burn period quickly drop back to just slightly above the
pre-burn values, with a mean of $6.35\times10^{-2}$ °C m s$^{-1}$ (64 W m$^{-2}$) and a standard deviation of
$3.76\times10^{-2}$ °C m s$^{-1}$(38 Wm$^{-2}$). However, the post-burn period contains several outliers (above the
99.3% percentile), indicating the influence of smoldering on some of the sonic anemometers
even after the fire has exited the burn plot. A specific example of the smoldering effect is the D4
sonic anemometer, where the post-burn $\overline{T'w'}$ (0.126 °C m s$^{-1}$ or 126 W m$^{-2}$) is about twice the
pre-burn value. The overall modest increase of $\overline{T'w'}$ in the post-burn period compared to the pre-
burn period was also observed in the two wildland fire experiments described in Heilman *et al.*

(2019).


**3.5 Quadrant Analyses**
**3.5.1 Turbulent heat fluxes**



The analysis above provided a quantitative assessment of fire-induced changes to the

turbulent heat and momentum fluxes through comparisons of flux values between the pre-burn
and the burn periods. However, such analysis cannot reveal what types of heat or momentum
transfer events are mostly affected by the fire. We apply the quadrant analysis method (also
known as sweep-ejection analysis) described earlier (Table 1) to the observed turbulent fluxes to
provide additional insight into how the fire changes the composition of heat and momentum
fluxes. By partitioning the total heat and momentum fluxes into four quadrants representing
different types of flux events, the quadrant or sweep-ejection analysis allows for the delineation
of the fire influence on specific types of turbulent heat and momentum transfer processes.

Figure 13 shows the relative contributions and the proportional number of occurrence of

the different heat-flux events (i.e., sweeps, ejections, outward interactions and inward
interactions) during each period, observed by each of the 16 sonic anemometers. During the pre-
burn period, the partitioning among the four types of events (see Table 1) by contribution and
proportion exhibits little variation across the 16 sonic anemometers. At all locations, the ejection
and sweep dominate, accounting for over 60% of the total events, with sweep being slightly
larger. The rest is split between outward interaction and inward interaction events, with the
former slightly outnumbering (20-23%) the latter (14-19%). A similar partitioning is observed
for the event contributions for the heat fluxes, but the ejection events, despite being slightly less
frequent, contribute more to the heat flux than do the sweep events. This apparent inconsistency
between the partitioning of the event number and the event contribution suggests that ejection
events likely involve larger eddies and stronger heat transfer compared to sweep events. This
pre-burn period partitioning is similar to previous ambient daytime measurements observed in
other studies (e.g., Heilman *et al.*, 2021).





The burn period is marked by substantial heterogeneity across the 16 sonic anemometers.
Despite differences in the magnitudes of contributions to the heat fluxes amongst the sonic
anemometers, the increases in the overall positive mean heat flux during the burn period can be
largely attributed to increases of ejection events that contribute to positive heat fluxes through
upward transfer of warmer air from the combustion zone to the atmosphere above. There is also
an increase in the negative contribution from inward interaction events, which represents the
downward transfer of warmer air from the atmosphere to the combustion zone. The contributions
to the overall mean heat flux by the other two types of events, sweep and outward interaction,
show little change from the pre-burn to the burn periods, which suggests that the turbulent heat
transfer processes represented by these types of events, namely downward transfer of colder air
from above to the surface or upward transfer of colder air from the combustion zone to the
atmosphere, are not very sensitive to the presence of a low-intensity fuel-bed-scale surface fire.
Compared to the partitioning in event contribution, the fire-induced changes to the
partitioning in event number are less clear. In general, the sonic anemometers that show an
increase in the contribution by inward interaction events also exhibit an increase in the number
of inward interaction events from the pre-burn to the burn periods. However, an increased
contribution to the overall mean heat flux by ejection events does not correspond to an increase
in the number of the ejection events. The increased number of sweep events are in agreement
with the increased sweep contributions at several sonics (A2-A4 and B2-B4), although the sweep
contributions are overwhelmed by that of the ejection contributions at these sonic anemometers.
A key finding from this heat flux sweep-ejection analysis is that turbulent heat fluxes
during the burn period are overwhelmingly dominated by ejection events, but there is usually a
small or no increase in the number of ejection events. This suggests that the presence of a low-

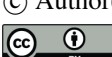



intensity fuel-bed-scale fire does not necessarily produce more upward turbulent heat transfer
events, but instead, it produces stronger events that quickly transfer and diffuse the sensible heat
generated by combustion into the ambient atmosphere above.

During the post-burn period, most sonic anemometers show vertical heat flux values that

are smaller than the burn period but still larger than the pre-burn period. The largest contribution
to the overall mean heat flux is usually from sweep events, accompanied also by an increase in
the number of the events, indicating the occurrence of many events where cold air is transferred
downward. The post-burn period also exhibits an increase in the heat-flux contributions from
outward interaction events, which represent downward transfer of warm air. Similar to the burn
period, inward interaction events, both in contribution and number, vary considerably across the
sonic array.

Figure 14 shows the partitioning of both the event number and the event contribution to

turbulent heat fluxes using data from all 16 sonic anemometers, which highlights more clearly
how the fire modifies the overall heat flux regime. Similar to the heat flux quadrant analysis for
individual sonic anemometers, the heat flux events averaged across the sonic anemometer array
for the pre-burn period is dominated by sweep (32%) and ejection (28%) events. Inward
interaction events occur with the least proportion (17%), followed by outward interaction events
(23%). The sweep and ejection events, which contribute to positive heat fluxes, are much larger
in magnitude than the negative heat flux contributions from the inward and outward interaction
events. The dominance of sweep and ejection events for the turbulent heat fluxes during the pre-
burn period follows observations made in previous studies (Heilman *et al.*, 2021).

The combined proportions of sweep and ejection events (both contributing to positive

heat fluxes) and the outward and inward interaction events (both contributing to negative heat

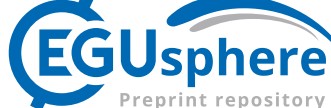

fluxes) remain similar between the burn and the pre-burn period. However, between the two
types of events in each group, one (sweep, inward interaction) increases and the other (ejection,
outward interaction) decreases in proportion. Previous fire experiments also reported an increase
in sweep events and a generally proportional decrease in ejection events (Heilman *et al.*, 2021),
but the magnitudes of the changes are larger than what is observed here, likely because the
previous fires are more intense. Additionally, modest changes in the partitioning of the event
number and contributions for this fire could be a byproduct of combining data from sonic
anemometers that are not strongly affected by the fire front (i.e. the westernmost sonic
anemometers) with those that experience more substantial changes.

The large changes in the contributions of the heat flux events during the burn period

suggest that this fire has greater impacts on the event contributions to the mean turbulent heat
fluxes than on the event number. Specifically, ejection event contributions dominate in the burn
period, making up 70.4% of the total contribution, while sweep and outward interaction
contributions decrease by a third and a sixth, respectively, compared to their contributions during
the pre-burn period. The magnitude of the contribution from inward interaction events increases
slightly but is quite similar to the contribution during the pre-burn period.

Heat flux events in the post-burn period more closely resemble the pre-burn period than

the burn period, but the event contributions and the event number do not return entirely to their
pre-burn values. As noted in the analyses of *TKE* and kinematic heat flux (Figures 9 and 11), this
result is consistent with smoldering occurring in the burn plot during the post-burn period. The
sweep event contribution during the post-burn period is 1.5 times higher than during the pre-burn
period and 1.3 times higher than during the burn period. Compared to the pre-burn values, the
post-burn period event contributions are slightly higher for outward interaction events and



slightly lower for ejection and inward interaction events. Overall, the post-burn period is
dominated by contributions from sweep events (37.7%), which is followed by ejection event
(25.3%) although lower than pre-burn values. These results differ somewhat from the Heilman *et*
*al.* (2021) in that they reported both sweep and ejection events returning to pre-burn values,
while only ejection events return to pre-burn values for this fire.

**3.5.2 Turbulent momentum fluxes**

Quadrant analysis is also applied to partition the vertical turbulent kinematic flux of

horizontal momentum $\overline{S'w'}$ into four different types and the results for each of the 16 sonic
anemometers are shown in Figure 15. During the pre-burn period, the overall mean momentum
fluxes are negative at all but two sonic anemometers (C1, C2) where the flux is slightly positive.
Between the two types of events that contribute to negative momentum fluxes, the sweep events
(downward transfer of higher horizontal momentum air from the atmosphere to the combustion
zone) contribute more than the ejection events (upward transfer of lower horizontal momentum
air from the combustion zone to the atmosphere above), which is consistent with the slightly
higher number of sweep events than ejection events. Between the two types of events that
contribute to positive momentum fluxes, the outward interaction events (upward transfer of
higher horizontal momentum air from the combustion zone to the atmosphere above) contribute
more than the inward interaction events (downward transfer of lower horizontal momentum air
from the atmosphere to the combustion zone), although the number of the inward and outward
interaction events is similar .



The changes from the pre-burn period to the burn period vary substantially by location,
but the sign of the overall mean momentum fluxes remains unchanged at most locations. The
most pronounced and consistent change across the anemometer array is a substantial increase in
the proportional number of inward interaction events and, to a lesser degree, the contribution
from these events. The ejection events also exhibit an increase in the number and the
contribution at most of the sonic anemometer locations. There is a general decrease in the
number of sweep and outward interaction events, but the contributions are not consistent, with
some sonic anemometers showing an increase while others experience a decrease in contribution.
An exception to the above general observations between the pre-burn and burn periods is
B4, where the overall momentum flux shifts from negative to positive due to an increase in
outward interaction contribution by as much as 5 times the pre-burn magnitude. The amount of
increase in the contribution from the outward interaction events, however, does not match the
small increase (approximately 10%) in the event number, which suggests that the increase in the
overall momentum flux magnitude at this location is likely due a small number of extremely
strong events of upward transfer of higher horizontal momentum air associated with large,
energetic eddies generated by the surface fire.
The large heterogeneity in the event contribution values for the momentum fluxes across
the sonic anemometer array during the burn period dissipated substantially into the post-burn
period. The event contribution and event number distributions once again become less dependent
on the locations of the sonic anemometers. Despite this tendency to return to the pre-burn
distribution, the post-burn period experiences larger contributions from, and higher number of
ejection and inward interaction events than sweep and outward interaction events, which is
opposite to the pre-burn period and similar to the burn period.



Figure 16 shows a quadrant analysis that combines data from all the sonic anemometers,

which allows for an assessment of how the fire modified the momentum flux turbulence regime

for the entire burn plot. Overall, sweep (31.9%) and outward interaction (26.6%) events

dominate the momentum flux contributions in the pre-burn period. The increases in the

proportion of inward interaction and ejection events from the pre-burn to the burn periods make

the contributions more balanced across the four quadrants, suggesting that the different event

contributions are more similar to each other during the burn than the pre-burn period. In the post-

fire period, inward interaction events contribute more to the mean momentum flux (25.7%) than

during the pre-fire period (18.1%). The event number distributions in the combined analysis

echoes the results from the individual sonic anemometers, with the pre-burn period showing

similar values for all four quadrants, a sharp increase in inward interaction events and decrease in

outward interaction events during the burn period, and fewer inward interaction events during the

post-burn period than during the burn period but more numerous than during the pre-burn period.

The results of the quadrant analysis of momentum fluxes presented above are somewhat

different from those of previous studies involving operational-scale prescribed burns. Heilman *et*

*al.* (2021) showed that during an intense grass fire and two low-intensity forest understory fires,

there can be substantial increase in the number and contribution of sweep and outward

interaction events and that the increase in the positive momentum flux from outward interaction

events largely offset the increase in the negative flux associated with sweep events. Whereas in

the small fuel-bed scale burn here, inward interactions occur most frequently, followed by

ejection events. However, the ejection event contributions to the mean momentum flux are larger

(32.3%), with the inward interaction event contributions (24.2%) more similar to the outward

interaction (23.4%) contributions. The feature of increased frequency of inward interaction



events and their increased contribution to the mean momentum flux compared to previous burns
is further observed in the post-burn period.

The event number and event contributions during the post-burn period also differ with

increased ejection and inward interactions events, 32.8% and 20.6%, while the large-scale burns
in Heilman *et al.* (2021) showed a closer return to pre-fire periods, with sweep and ejection
events making up the majority of event number and contributions. The contributions from sweep,
inward interaction, and ejection events remain elevated during the post-burn period, while the
contributions from outward interaction decrease during post-burn to values lower than the values
of the pre-burn period.

**4. Summary**

This study presents atmospheric turbulence observed using a 4 × 4 array of fast-response

3D sonic anemometers during a low-intensity fire experiment on a 10 m x 10 m burn plot in the
Silas Little Experimental Forest in New Jersey, USA. The density of turbulence measurements is
unprecedented for fire experiments, allowing for a deeper analysis of heterogeneities as the
surface line-fire spread through the burn plot than was previously possible. The analysis focuses
on assessments of the fire impacts on turbulence intensity, as measured by *TKE*, turbulent
momentum flux or shear stress as measured by friction velocity, and turbulent heat flux.

The influence of the low-intensity surface line-fire on the atmosphere above the

combustion zone is evidenced by an increase in temperature up to 20 ºC, the generation of strong
updrafts up to 6 m s$^{-1}$ and downdrafts up to -5 m s$^{-1}$ and a decrease in the streamwise velocity
coupled with an increase in the cross-stream velocity indicting horizontal convergence in the



vicinity of the fire front. The observed fire exhibited behavior more consistent with a buoyant
plume than mechanically forced rising motion resulting from converging surface air. The
influence of the fire on horizontal velocity components persisted longer after fire front passage
while the influence on vertical velocity subsided rapidly behind the fire front.

The fire modified turbulence characteristics at the fuel bed-atmosphere interface. There

was an increase in the turbulence intensity, with *TKE* values 2-3 times higher than the ambient
environment, due primarily to the increase in cross-stream velocity variance and, to a lesser
degree, the increase in the vertical velocity and streamwise velocity variance. Heilman *et al.*
(2017) also reported two to threefold increases in *TKE* values during two operational-scale low-
intensity forest understory prescribed fires. It is interesting to note that this increase in *TKE* is
only slightly smaller than what was observed during the intense grass fire during FireFlux
(Clements *et al.*, 2007), although the magnitude of *TKE* of the intense grass fire is substantially
larger than that of the low-intensity fires. Despite this increase in *TKE*, the value of *TKE* was still
smaller than what is expected in an environment of substantial turbulence. Additionally, despite
the increase in the vertical velocity variance during the fire, the *TKE* was still dominated by the
horizontal velocity variance, indicating that the turbulence regime remained anisotropic
(anisotropic ratio << 1/3) above the combustion zone of this low-intensity fuel-bed-scale surface
fire.

The fire enhanced upward sensible heat fluxes substantially by as much as 40 times the

flux in the ambient atmosphere (from 50 W m$^{-2}$ to 2 kW m$^{-2}$). This change in the sensible heat
flux is largely attributable to an increased contribution of upward transfer by turbulent eddies of
warmer air from the combustion zone to the atmosphere above, which is also known as ejection
events for vertical turbulent heat transfer. This increase in the contribution of the ejection events



to turbulent heat fluxes was not caused by a corresponding increase in the number of ejection
events that changed little from the pre-burn to burn periods. This mismatch between the ejection
event contribution and event number suggests that the presence of a low-intensity fuel-bed-scale
fire may not necessarily produce more upward turbulent heat transfer events, but rather, it can
produce strong ejection events associated with large, energetic eddies. The warmer air
transported upward by the ejection events can also be transported downward by inward
interaction events, which also increased somewhat during the fire.

Compared to the turbulent heat flux, the impact of the fire on turbulent momentum flux

or shear stress was less pronounced. In general, an increase in momentum fluxes was observed
during the burn, with friction velocity, a measure of total shear stress on horizontal wind, 2-3
times the ambient value (from ~ 0.25 ms$^{-1}$ to 0.6 ms$^{-1}$). Previous studies of operational-scale
grass fire or forest understory fires also found up to a 3-fold increase in friction velocity despite
that the scale of this fire is much smaller than the previous fires and that the absolute values of
friction velocity during the intense grass fire were 5 times higher than the low-intensity fire here
(Clements *et al.*, 2007; Heilman *et al.,* 2017; 2021). The fire was accompanied by an increase in
the downward transfer of lower horizontal momentum air, also known as inward interaction
events, along with a smaller increase in the upward transfer of lower horizontal momentum air
referred to as ejection events. This finding differs from previous observations during an
operational-scale forest understory fire where an increase in sweep (downward transfer of higher
horizontal momentum air) and outward interaction (upward transfer of higher horizontal
momentum air) contributions to the mean momentum fluxes were detected (Heilman *et al.*,

2021).



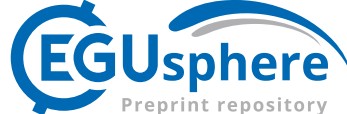

Perhaps the most significant finding from this study is the large variations in the observed

fire-induced perturbations across the sonic anemometer array in the burn plot. The anemometers

on the western side of the burn plot where a surface line-fire was ignited picked up very weak or

no signals of the fire despite the proximity to the initial fire line. In contrast, the sonic

anemometers in the center or eastern side of the burn plot picked up clear fire signals. Although

the features of fire-induced turbulence regime (e.g., anisotropy, sweep-ejection dynamics)

revealed by the sonic anemometers are similar, the magnitudes vary with downwind distance and

the relative position of the sonic anemometers to the impinging fire front. Considering the size of

the burn plot (10 m x 10 m) and the homogeneity of consumed fuels, this finding suggests that

considerable care should be taken when comparing, contrasting, and combining data from

multiple fires or from multiple instruments on the same fire to ensure that significant fire signals

are not being over- or under-represented in the analyses that inform the conclusions of the

studies. This also calls into question of using numerical simulations from coupled atmosphere-

fire behavior models with horizontal grid spacing $\geq 10$ m. The results presented here suggest that

1-2 m grid spacing is necessary for model simulations to capture atmospheric turbulent

circulations that have spatiotemporal scales similar to the scales associated with flame dynamics

in the combustion zone.

Future work will compare results from this case with those of other burns in the 10 m x

10 m burn series to delineate the effect of fuel and ambient atmospheric conditions on fire-

atmosphere interactions and with results from other prescribed-fire experiments to help scale up

or scale down the results between small-scale and operational scale fires. Future work will also

include the reanalysis of 10 Hz sonic anemometer data from other fire experiments using some

or all of the methodologies employed here, which could contribute to the identification and



documentation of a series of steps, protocols, standards, and methodologies by which 10-Hz
sonic anemometer data collected during fire experiments can be compared and contextualized.
Additionally, the data collected from the other instruments deployed during the SERDP fuel-bed-
scale fire experiments should be included in future analyses. Spectral and co-spectral analyses
should be performed to help understand the temporal and spatial scale of turbulence regimes at
the fuel-bed and atmosphere interface.

Because the burn period was chosen to be between the time when the first and the last

sonic anemometers have temperatures satisfying the threshold value (eight standard deviations in
these analyses), the burn period included time after the fire has passed the sonic anemometer
location, which likely yielded an underestimation of the fire effect. Similarly, the inclusion of all
16 sonic anemometers in the analysis, including those that registered little fire signal, likely
contributed to an underestimation. Consequently, fire-induced turbulent circulations and the
associated turbulent heat and momentum fluxes are likely to be stronger than what has been
reported here.

**Acknowledgements**
Founding for this project was provided by the U.S. Department of Defense Strategic
Environmental Research and Development (SERDP) program (Project Number: RC-2461). We
would like to acknowledge Jon Horm, Seoung-kyun Im, Robert Kremens, William Mell and
Albert Simeoni for their contributions to the original research proposal. We thank Zach
Campbell-Lochrie and Carlos Walker-Ravena for their help in the experiment design and
instrument deployment of the 10 m x 10 m burn series.




**Code and Data Availability**

Python language was used for all analyses and data management, with the Pandas package

(https://zenodo.org/record/7037953#.Yw-at3bMIp4) used for data processing, NumPy package

(https://numpy.org/) used for most statistical calculations and Matplotlib visualization package

(https://matplotlib.org/) used for plotting, , all of which are open source packaged in the Python

environment. The computer codes and the data are hosted on software sharing and version

control website and service GitHub. https://github.com/JosephSeitz/SERDP-10x10meter-Burn-

Cleaner.

All data used in this study are publicly archived and available via the USFS Data Archive (in

press, links to be included in revised version).

**Author Contributions**

All authors contributed to the research design. K.C., N.S., M.G., M.P., R.H. and E.M. conducted

the fire experiment and collected the data. J.C. and M.P., with assistance from K.C., did the

initial process and formatting of the data. J.S., with assistance and guidance from J.J.C. and

discussions and feedback from S.Z., W.H., X.B. M.K., performed the data analysis and produced

all the plots. S.Z. wrote the manuscript. M.G., W.H., K.C. and N.S. edited the manuscript.





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

Clark, Kenneth L.; Gallagher, Michael R.; Mueller, Eric V.; Hadden, Rory M.; Walker-Ravena,
Carlos; Campbell-Lochrie, Zakary J.; Cole, Jason A.; Patterson, Matthew M.; Everland,
Alexis I.; Skowronski, Nicholas S. 2022a. Multi-scale analyses of wildland fire combustion
processes: Small-scale field experiments - three-dimensional wind and temperature. Fort
Collins, CO: Forest Service Research Data Archive.

Clark, Kenneth L.; Gallagher, Michael R.; Mueller, Eric V.; Hadden, Rory M.; Walker-Ravena,
Carlos; Campbell-Lochrie, Zakary J.; Cole, Jason A.; Patterson, Matthew M.; Everland,
Alexis I.; Skowronski, Nicholas S. 2022b. Multi-scale analyses of wildland fire combustion
processes: Small-scale field experiments - temperature profile. Fort Collins, CO: Forest
Service Research Data Archive.

Clark, K.L., Heilman, W.E., Skowronski, N.S., Gallagher, M.R., Mueller, E., Hadden, R.M., and
Simeoni, A. (2020) Fire behavior, fuel consumption, and turbulence and energy exchange
during prescribed fires in pitch pine forests. *Atmosphere,* **11**, 242.

Clark, T.L., Jenkins, M.A., Coen, J.L. and Packham, D.R., (1996) A coupled atmosphere-fire
model: Role of the convective Froude number and dynamic fingering at the fireline.
*International Journal of Wildland Fire*, **6**, pp.177-190.

Clements, C.B., and Seto, D. (2015) Observations of fire-atmosphere interactions and near-
surface heat transport on a slope. *Boundary-Layer Meteorology,* **154**, 409-426.



Clements, C.B., Kochanski, A.K., Seto, D., Davis, B., Camacho, C., Lareau, N.P., Contezac, J., Restaino, J., Heilman, W.E., Krueger, S.K. and Butler, B. (2019) The FireFlux II experiment: a model-guided field experiment to improve understanding of fire–atmosphere interactions and fire spread. *International Journal of Wildland Fire*, **28**, 308-326.

Clements, C.B., Kochanski, A.K., Seto, D., Davis, B., Camacho, C., Lareau, N.P., Contezac, J., Restaino, J., Heilman, W.E., Krueger, S.K., Butler, B., Ottmar, R.D., Vihnanek, R., Flynn, J., Filippi, J.B., Barboni, T., Hall, D.E., Mandel, J., Jenkins, M.A., O'Brien, J., Hornsby, B., and Teske, C. (2019) The FireFlux II experiment: a model-guided field experiment to improve understanding of fire–atmosphere interactions and fire spread. *International Journal of Wildland Fire,* **28,** 308-326.

Clements, C.B., Zhong, S., Bian, X., Heilman, W.E., and Byun, D.W. (2008), First observations of turbulence generated by grass fires. *Journal of Geophysical Research*, **113**, D22102.

Clements, C.B., Zhong, S., Goodrick, S., Li, J., Potter, B.E., Bian, X., Heilman, W.E., Charney, J.J., Perna, R., Jang, M. and Lee, D. (2007) Observing the dynamics of wildland grass fires: FireFlux—A field validation experiment. *Bulletin of the American Meteorological Society*, **88**, 1369-1382.

Di. Christina, G., Gallagher, M., Skowonski, N., Simeoni, In, S.-K. (2022 Design and implementation of a portable large-scale wind tunnel for wildfire research. *Fire Safety Journal*, **131**, 103607.

Ebel, B.A. (2013) Simulated unsaturated flow processes after wildfire and interactions with slope aspect. *Water Resources Research,* **49**, 8090-8107



Finney, M.A., Cohen, J.D., Forthofer, J.M., McAllister, S.S., Golner, M.J., Gorham, D.J., Saito, K., Akafuah, N.K., Adam, B.A., and English, J.D. (2015) Role of buoyant flame dynamics in wildfire spread. *Proceedings of the National Academy of Sciences,* **112**, 9833-9838.

Forthofer, J.M., and Goodrick, S.L. (2011) Review of vortices in wildland fire. *Journal of Combustion,* **2011**, Article ID 984363.

Gallagher, Michael R.; Skowronski, Nicholas S.; Hadden, Rory M.; Mueller, Eric V.; Clark, Kenneth L.; Campbell-Lochrie, Zakary J.; Walker-Ravena, Carlos; Kremens, Robert L.; Everland, Alexis I.; Patterson, Matthew M.; Cole, Jason A.; Heilman, Warren E.; Charney, Joseph J.; Bian, Xindi; Mell, William E.; Hom, John L.; Im, Seong-kyun; Kiefer, Michael T.; Zhong, Shiyuan; Simeoni, Albert J.; Rangwala, Ali; Di Cristina, Giovanni. (2022) Multi-scale analyses of wildland fire combustion processes: Small-scale field experiments – plot layout and documentation. Fort Collins, CO: Forest Service Research Data Archive

Heilman, W.E. (2021) Atmospheric turbulence in wildland fire environments: implications for fire behavior and smoke dispersion. *Fire Management Today. 79*, pp.24-29.

Heilman, W.E. and Bian, X. (2013) Climate variability of near surface turbulent kinetic energy over the United States: Implications for fire weather prediction. *Journal of Applied Meteorology and Climatology*, 52, 753-772.

Heilman, W.E., Barnerjee, T., Clements, C.B., Clark, K.L., Zhong, S., and Bian X. (2021) Observations of sweep-ejection dynamics for heat and momentum fluxes during wildland fires in forested and grassland environments. *Journal of Applied Meteorology and Climatology*, **60**, 185-199.



Heilman, W.E., Bian, X., Clark, K.L. and Zhong, S. (2019) Observations of turbulent heat and
    momentum fluxes during wildland fires in forested environments. *Journal of Applied
    Meteorology and Climatology*, **58**, pp.813-829.

Heilman, W.E., Bian, X., Clark, K.L., Skowronski, N.S., Hom, J.L. and Gallagher, M.R. (2017)
    Atmospheric turbulence observations in the vicinity of surface fires in forested
    environments. *Journal of Applied Meteorology and Climatology*, **56**, 3133-3150.

Heilman, W.E., Clements, C.B., Seto, D., Clark, K.L., Skowonski, N.S., and Hom, L.J. (2015)
    Observations of fire-induced turbulence regimes during low-intensity wildland fires in
    forested environments: Implications for smoke dispersion. *Atmospheric Science Letters*, **16**,
    453-460.

Heilman W.E. and Bian, X. (2013)

Katul, G., Poggi, D., Cava, D., and Finnigan, J. (2006) The relative importance of ejections and
    sweeps to momentum transfer in the atmospheric boundary layer. *Bound.-Layer Meteor.*,
    **120**, 367–375.

Katul, G., Kuhn, G., Schieldge, J., and Hsieh, C.-I. (1997) The ejection sweep character of scalar
    fluxes in the unstable surface layer. *Bound.-Layer Meteor.*, **83**, 1–26.

Kitzberger, T., Falk, D.A., Westerling, A.L., and Swetnam T.W. (2017) Direct and indirect
    climate controls predict heterogeneous early-mid 21st century wildfire burned area across
    western and boreal North America. *PLOS ONE,* **12**, e0188486.

Kremens, Robert L.; Gallagher, Michael R.; Clark, Kenneth L.; Mueller, Eric V.; Hadden, Rory
    M.; Heilman, Warren E.; Charney, Joseph J.; Hom, John L.; Campbell-Lochrie, Zakary J.;



Walker-Ravena, Carlos; Everland, Alexis I.; Cole, Jason A.; Patterson, Matthew M.; Skowronski, Nicholas S. 2022. Multi-scale analyses of wildland fire combustion processes: Small-scale field experiments - fire radiative power. Fort Collins, CO: Forest Service Research Data Archive.

Littell, J.S., Peterson, D.L., Riley, K.L., Liu, Y. and Luce, C.H. (2016). A review of the relationships between drought and forest fire in the United States. *Global Change Biology, 22*, 2353-2369.

Moody, M.J., Gibbs, J.A., Kruger, S., Mallia, D., Pardyjak, E.R., Kochanski, A.K., Bailey, B.N., Stoll, R. (2022) QES-Fire, a dynamically coupled fast-response wildfire model. International Journal of Wildland Fire, **31,** 306-325.

Mueller, E.V., Skowronski, N.,Clark, K., Gallagher, M., Kremens, R., Thomas, J.C., El Houssami, M., Filkov, A., Hadden, R.M., Mell, W.; et al. (2017) Utilization of remote sensing techniques for the quantification of fire behavior in two pine stands. *Fire Safety Journal*, **91**, 845–854, doi:10.1016/j.firesaf.2017.03.076.

Potter, B.E. (1996) Atmospheric properties associated with large wildfires. *International Journal of Wildland Fire* **6**, 71–76.

Potter, B.E. (2012): Atmospheric interactions with wildland fire behavior – I: Basic surface interactions, vertical profiles and synoptic structures. *International Journal of Wildland Fire*, **21**, 779-801.

Povak, N.A., Hessburg, P.F. and Salter, R.B. (2018) Evidence for scale-dependent topographic controls on wildfire spread. *Ecosphere,* **9**(10): e02443.



Seto, D., Strand, T.M., Clements, C.B., Thistle, H., and Mickler, R. (2014) Wind and plume thermodynamic structures during low-intensity subcanopy fires. *Agricultural and Forest Meteorology,* **198-199**, 53-61.

Seto D., Clements, C.B., and Heilman, W.E. (2013) Turbulence spectra measured during fire front passage. *Agricultural and Forest Meteorology,* **169**, 195-210.

Seto, D., and Clements, C.B. (2011) Fire whirl evolution observed during a valley wind-sea breeze reversal. *Journal of Combustion*, **2011**, 12pp https://doi.org/10.1155/2011/569475

Sharples, J.J. (2009) An overview of mountain meteorological effects relevant to fire behaviour and bushfire risk. *International Journal of Wildland Fire* **18**, 737-754.

Sharples, J.J., McRae, R.H.D., Wilkes, S.R. (2012) Wind–terrain effects on the propagation of wildfires in rugged terrain: Fire channelling. *International Journal of Wildland Fire,* **21**, 282-296.

Skowronski, N.S. (2021) Multi-scale analysis of wildland fire combustion processes in open canopy forests using coupled iteratively informed laboratory-, field- and mode-based approach. Final Technical Report, SERDP Project RC-2641. Available at https://www.serdp-estcp.org/Program-Areas/Resource-Conservation-and-Resiliency/Air-Quality/Fire-Emissions/RC-2641

Skowronski, Nicholas S.; Charney, Joseph J; Clark, Kenneth L.; Gallagher, Michael R.; Hadden, Rory M.; Heilman, Warren E.; Hom, John L.; Kremens, Robert L.; Cole, Jason A.; Campbell-Lochrie, Zakary J.; Walker-Ravena, Carlos; Mueller, Eric V.; Everland, Alexis I.; Patterson, Matthew M. 2022a. Multi-scale analyses of wildland fire combustion



processes: Small-scale field experiments - infrared data. Fort Collins, CO: Forest Service
Research Data Archive.

Skowronski, Nicholas S.; Charney, Joseph J; Clark, Kenneth L.; Gallagher, Michael R.; Hadden,
Rory M.; Heilman, Warren E.; Hom, John L.; Kremens, Robert L.; Cole, Jason A.;
Campbell-Lochrie, Zakary J.; Walker-Ravena, Carlos; Mueller, Eric V.; Everland, Alexis
I.; Patterson, Matthew M. 2022b. Multi-scale analyses of wildland fire combustion
processes: Small-scale field experiments – terrestrial laser scans. Fort Collins, CO: Forest
Service Research Data Archive.

Skowronski, N., and Hom, J.L. (2015): Observations of fire-induced turbulence regimes during
low-intensity wildland fires in forested environments: Implications for smoke dispersion.
*Atmospheric Sciences Letters*, **16**, 453–460.

Stull, R.B., (1988) An introduction to boundary layer meteorology (Vol. 13). Springer Science &
Business Media.

Viegas, D.X., and Neto, L.P. (1991) Wall shear stress as a parameter to correlate the rate of
spread of a wind-induced forest fire. *International Journal of Wildland Fire,* **1**, 177–188.

Werth, P.A., Potter, B.E., Clements, C.B., Finney, M.A., Goodrick, S.L., Alexander, M.E., Cruz,
M.G., Forthofer, J.A., and McAllister, S.S. (2011) Synthesis of knowledge of extreme fire
behavior: For fire managers. General Technical Report PNW-GTR-854, US Department of
Agriculture, Forest Service, Pacific Northwest Research Station, Vol. I. Portland, OR, 144.



Table 1. Vertical turbulent transfer events and the associated quadrat designations.

| Q | $\varphi'w'$ | $\varphi'$ | $w'$ | Heat flux | Momentum flux |
|---|---|---|---|---|---|
| 1 | >0 | >0 | >0 | Ejection: upward flux of warmer air | Outward Interaction: upward flux of lower horizontal momentum air |
| 2 | <0 | <0 | >0 | Inward Interaction: downward flux of warmer air | Sweep: downward flux of higher horizontal momentum air |
| 3 | >0 | <0 | <0 | Sweep: downward flux of cooler air | Inward Interaction: downward flux of lower horizontal momentum air |
| 4 | <0 | <0 | >0 | Outward Interaction: upward flux of cooler air | Ejection: upward flux of higher horizontal momentum air |



**LIST OF FIGURES**

Figure 1.  Sketch of the burn plot and the instruments deployed to the plot. The four capital letters (A, B, C and D) denote the four trusses and the four numbers (1, 2, 3, 4) refer to the 3D sonic anemometers on the trusses.  Posts hanging on trusses B and C show the heights and location of thermocouples. The center post indicates the position of the infrared camera. The boxes next to the sonic anemometers indicate the radiometer/spectral camera pairs. The rectangular box on the ground indicates fuel cells for fuel loading estimation.  The symbol near B2 indicates the TACO for emission data collection

Figure 2.  Surface meteorological condition on May 20, 2019, the day of the experimental burn, observed by the weather station approximately 200 m northeast of the burn plot.

Figure 3. Infrared images taken at 10 m above the center of the burn plot showing fuel bed temperature before a), near b) and after c-f) ignition.

Figure 4. Time series of 10-Hz observations of temperature ($T$), horizontal wind speed ($S$) and vertical wind component ($w$) observed by the 16 sonic anemometers.

Figure 5. The number of sonic anemometers that recorded temperatures at or above a given threshold value (left) and the length of period over which the threshold was reached or exceeded (right). The symbol $\sigma$ denotes pre-burn period temperature standard deviation.

Figure 6. Time series of 10 Hz streamwise ($u$, blue) and cross-stream ($v$, green) wind velocity components and temperature perturbations ($T'$, red) recorded by each sonic anemometer at 2.5 m above the ground. The vertical dashed black lines indicate the burn period determined by the first and last occurrence of $T' \geq 8\sigma$. Time is the minutes since the start of the pre-burn period.

Figure 7. Distributions of 10 Hz streamwise ($u$), cross-stream ($v$), and vertical ($w$) wind velocity components, and temperature perturbations ($T'$) from all 16 sonic anemometers during pre-burn, burn and post-burn periods. The box represents the $25^{th}$ and $75^{th}$ percentile of the data, with data inside the whiskers representing $99.3\%$ of the data. The orange line in the boxes is the median value, the green triangle is the mean, and the blue shading is the density of values of the data.

Figure 8. Time series of 1-minute averaged turbulent kinetic energy ($TKE$) (red) for each sonic anemometer and the three components of velocity variance, $u'^2/2$ (yellow), $v'^2/2$ (blue) and $w'^2/2$ (green), that make up the $TKE$. The vertical dashed black lines indicate the burn period determined by the first and last occurrence of $T' \geq 8\sigma$. Time is the minutes since the start of the pre-burn period.

Figure 9. Distributions of turbulent kinetic energy ($TKE$) and the three components of velocity variance ($u'^2/2$, $v'^2/2$ and $w'^2/2$) that make up the $TKE$ from all 16 sonic anemometers during the pre-burn, burn and post-burn periods. The box represents the $25^{th}$ and $75^{th}$ percentile of the data, with data inside the whiskers representing $99.3\%$ of the data. The orange line in the boxes is the median value, the green triangle is the mean, and the blue shading is the density of values of the data.



Figure 10. Time series of 1-minute averaged friction velocity squared ($u_*{}^2$, pink pluses) and its two components, the streamwise kinematic momentum flux, $\overline{u'w'}$ (yellow circle) and the cross-stream kinematic momentum flux, $\overline{v'w'}$ (blue diamonds), for each of the 16 sonic anemometers. The vertical dashed black lines indicate the burn period determined by the first and last occurrence of $T' \geq 8\sigma$. Time is the minutes since the start of the pre-burn period.

Figure 11. Distributions of friction velocity squared ($u_*{}^2$) and its two components ($\overline{u'w'}$ and $\overline{v'w'}$) from all 16 sonic anemometers during the pre-burn, burn, and post-burn periods. The box represents the *25th* and *75th* percentile of the data, with data inside the whiskers representing *99.3%* of the data. The orange line in the boxes is the median value, the green triangle is the mean, and the blue shading is the density of values of the data.

Figure 12. Time series of 1-minute averaged heat flux for each of the 16 sonic anemometers (left) and the distribution of heat fluxes from all 16 sonic anemometers during the pre-burn, burn, and post-burn periods (right). The box represents the *25th* and *75th* percentile of the data, with data inside the whiskers representing *99.3%* of the data. The orange line in the boxes is the median value, the green triangle is the mean, and the blue shading is the density of values of the data.

Figure 13. Quadrant analysis of the instantaneous vertical kinematic turbulent heat fluxes showing the contributions to the total flux from (top row), and the percent of (bottom row) the four types of events: outward interaction (green), ejection (red), inward interaction (blue), and sweep (orange) for each of the 16 sonic anemometers during the pre-burn, burn, and post-burn periods. The black diamonds in the top row indicate the total heat flux values. The sonic anemometers are arranged from west to east roughly following the fire spread across the burn plot.

Figure 14. Quadrant analysis of the instantaneous vertical kinematic turbulent heat fluxes showing the contributions to the total flux from (top row), and the percent of (bottom row) the four types of events: outward interaction (green), ejection (red), inward interaction (blue), and sweep (orange) for all 16 sonic anemometers during the pre-burn, burn, and post-burn periods. The black diamonds in the top row indicate the total heat flux values. The sonic anemometers are arranged from west to east roughly following the fire spread across the burn plot.

Figure 15. Quadrant analysis of the instantaneous vertical kinematic turbulent fluxes of horizontal momentum showing the contributions to the total flux from (top row), and the percent of (bottom row) the four types of events: outward interaction (red), sweep (green), inward interaction (orange), and ejection (blue) for each of the 16 sonic anemometers during the pre-burn, burn, and post-burn periods. The black diamonds in the top row indicate the total flux values. The sonic anemometers are arranged from west to east roughly following the fire spread across the burn plot.

Figure 16. Quadrant analysis of the instantaneous vertical kinematic turbulent fluxes of horizontal momentum showing the contributions to the total flux from (top row), and the percent of (bottom row) the four types of events: outward interaction (red), sweep (green), inward





interaction (orange), and ejection (blue) for all 16 sonic anemometers during the pre-burn, burn, and post-burn periods. The black diamonds in the top row indicate the total flux values. The sonic anemometers are arranged from west to east roughly following the fire spread across the burn plot.



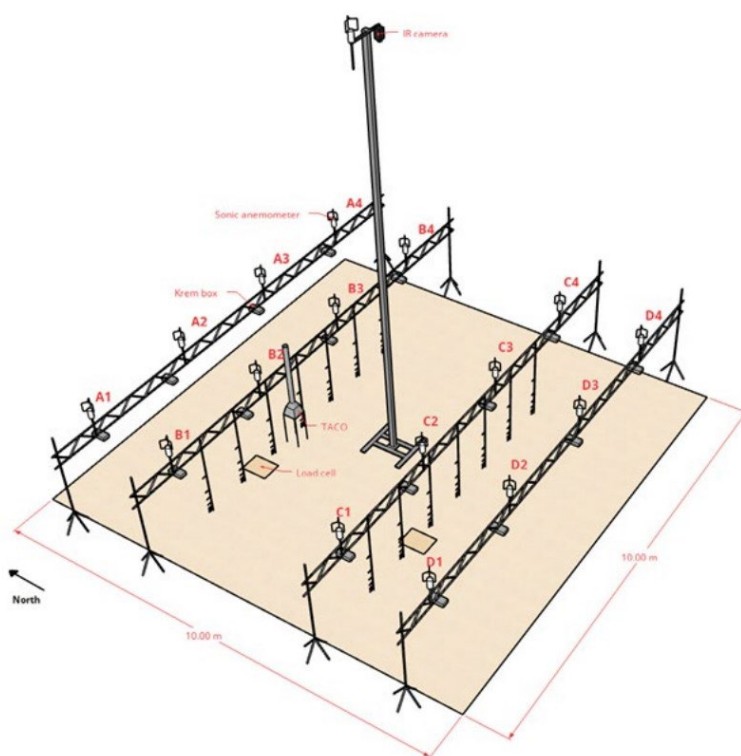

Figure 1.  Sketch of the burn plot and the instruments deployed to the plot. The four capital letters (A, B, C and D) denote the four trusses and the four numbers (1, 2, 3, 4) refer to the 3D sonic anemometers on the trusses. Posts hanging on trusses B and C show the heights and location of thermocouples. The center post indicates the position of the infrared camera. The boxes next to the sonic anemometers indicate the radiometer/spectral camera pairs. The rectangular box on the ground indicates fuel cells for fuel loading estimation. The symbol near B2 indicates the TACO for emission data collection



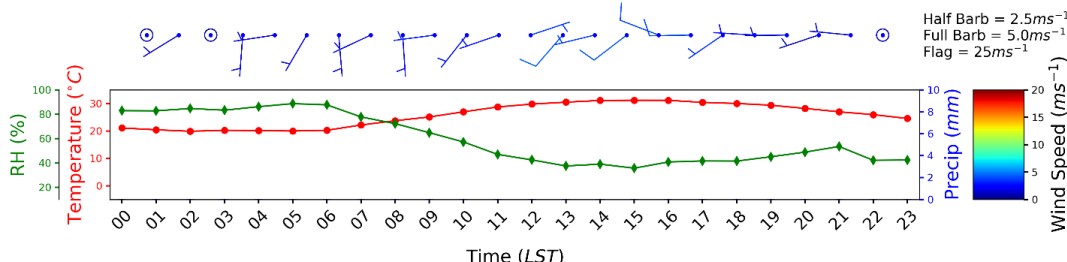

Figure 2. Surface meteorological condition on May 20, 2019, the day of the experimental burn, observed by the weather station approximately 200 m northeast of the burn plot.





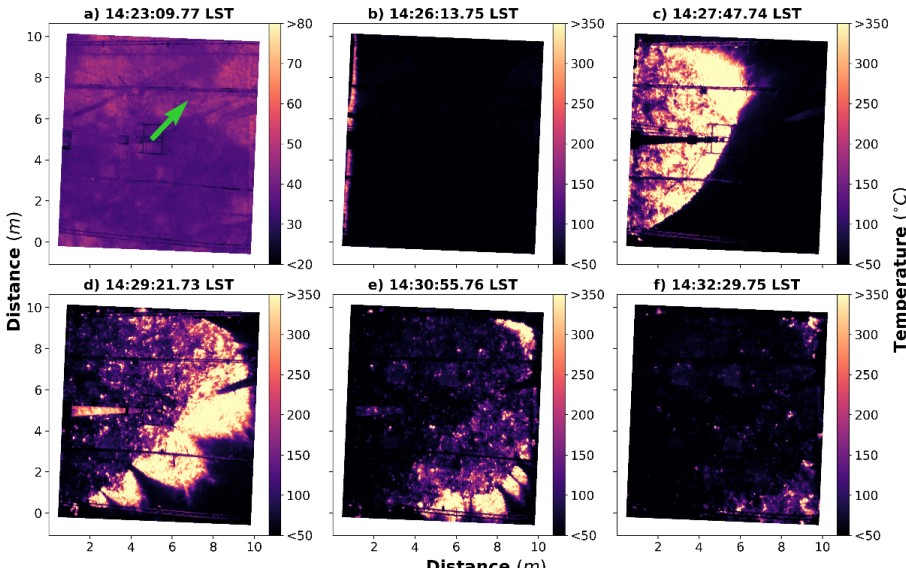

Figure 3. Infrared images taken at 10 m above the center of the burn plot showing fuel bed temperature before a), near b) and after c-f) ignition.

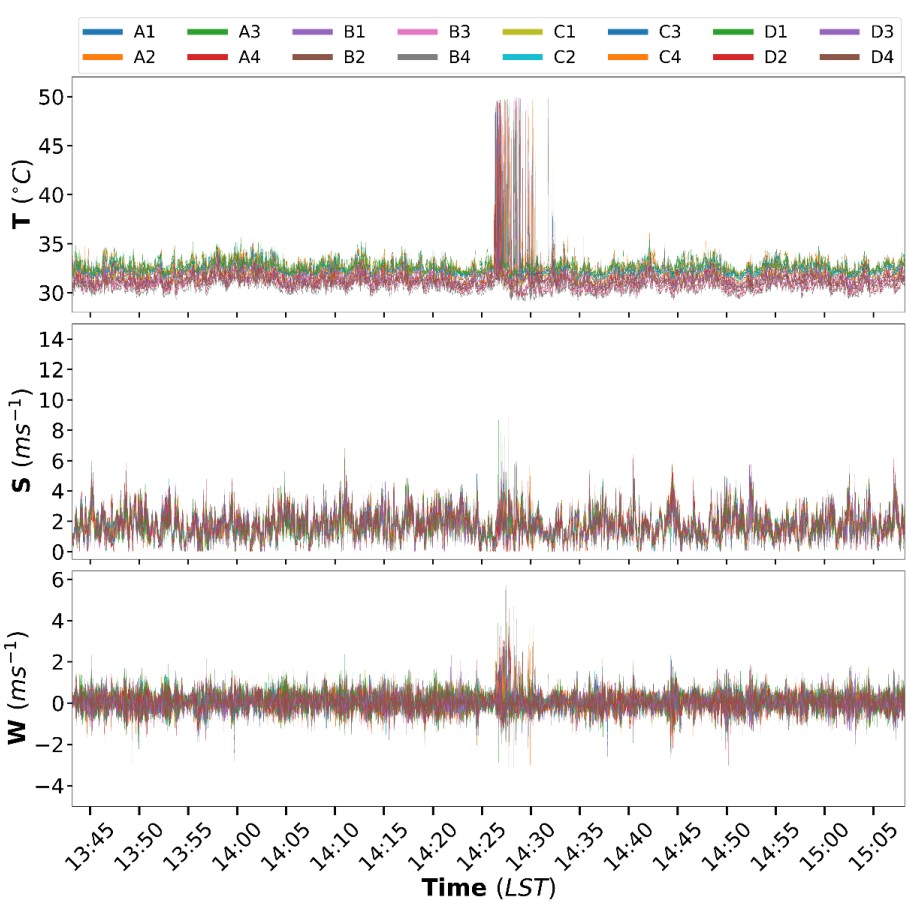

Figure 4. Time series of 10-Hz observations of temperature ($T$), horizontal wind speed ($S$) and vertical wind component ($w$) observed by the 16 sonic anemometers.



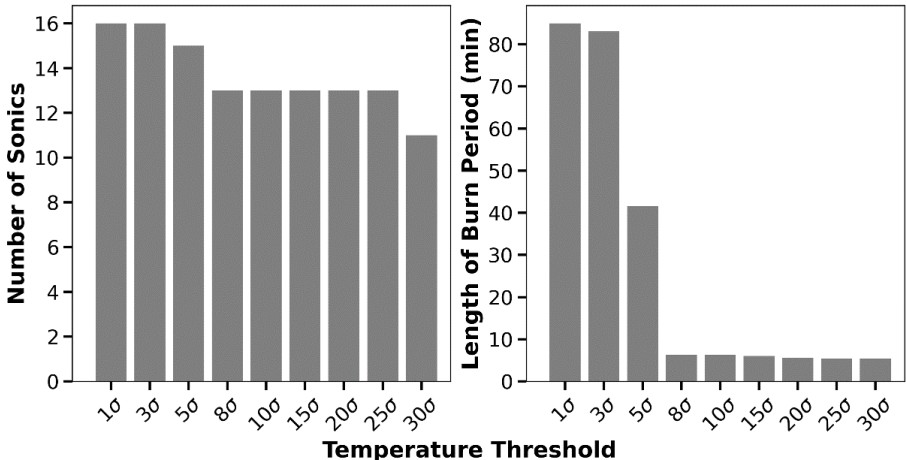

Figure 5. The number of sonic anemometers that recorded temperatures at or above a given threshold value (left) and the length of period over which the threshold was reached or exceeded (right). The symbol $\sigma$ denotes pre-burn period temperature standard deviation.



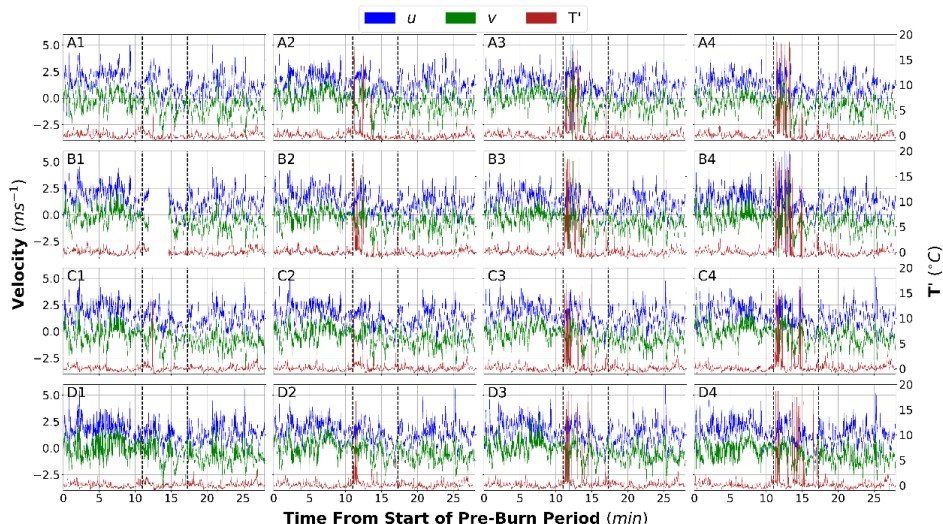

Figure 6. Time series of 10 Hz streamwise (*u*, blue) and cross-stream (*v*, green) wind velocity components and temperature perturbations (*T'*, red) recorded by each sonic anemometer at 2.5 m above the ground. The vertical dashed black lines indicate the burn period determined by the first and last occurrence of $T' \geq 8\sigma$. Time is the minutes since the start of the pre-burn period.





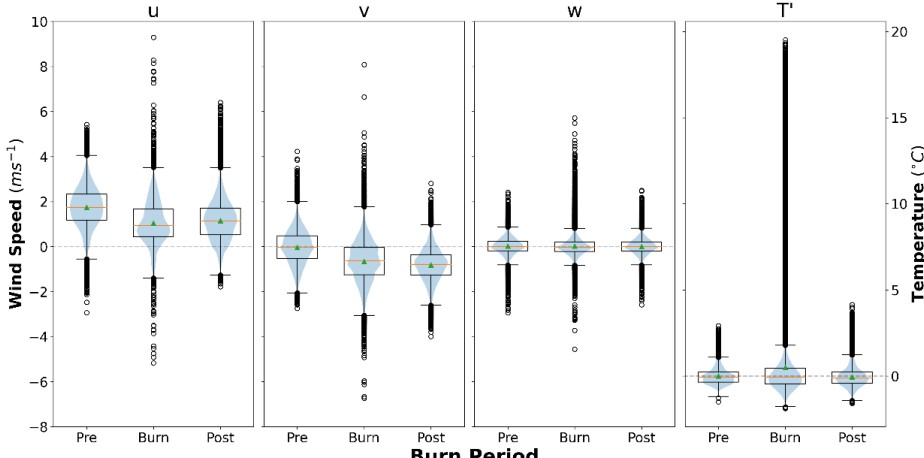

Figure 7. Distributions of 10 Hz streamwise ($u$), cross-stream ($v$), and vertical ($w$) wind velocity components, and temperature perturbations ($T'$) from all 16 sonic anemometers during pre-burn, burn and post-burn periods. The box represents the $25^{th}$ and $75^{th}$ percentile of the data, with data inside the whiskers representing $99.3\%$ of the data. The orange line in the boxes is the median value, the green triangle is the mean, and the blue shading is the density of values of the data.





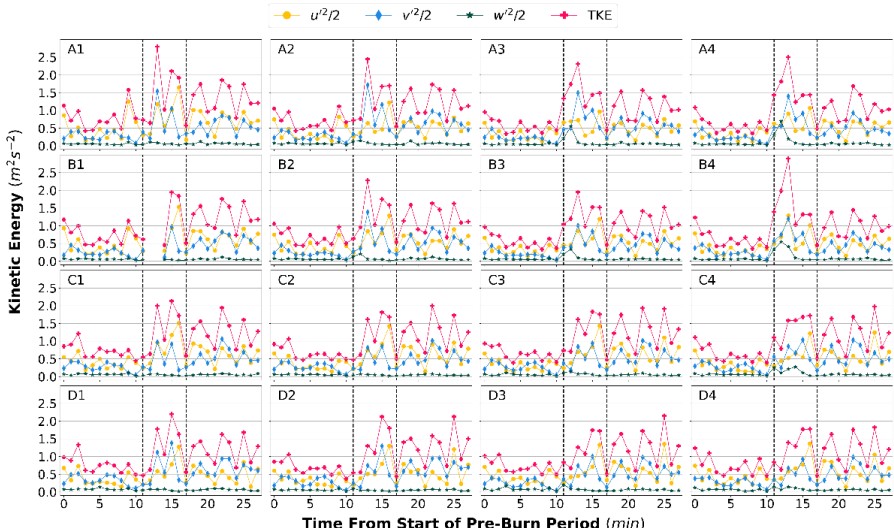

Figure 8. Time series of 1-minute averaged turbulent kinetic energy (*TKE*) (red) for each sonic anemometer and the three components of velocity variance, $u'^2/2$ (yellow), $v'^2/2$ (blue) and $w'^2/2$ (green), that make up the *TKE*. The vertical dashed black lines indicate the burn period determined by the first and last occurrence of $T' \geq 8\sigma$. Time is the minutes since the start of the pre-burn period

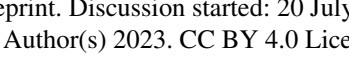



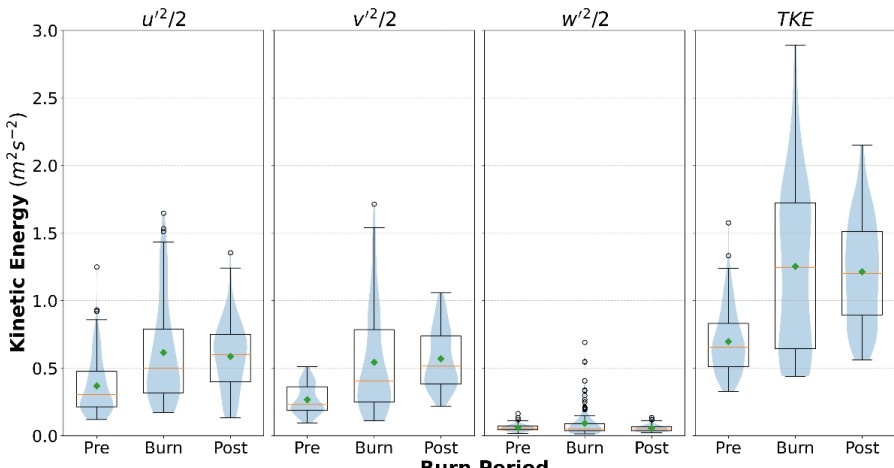

Figure 9. Distributions of turbulent kinetic energy (*TKE*) and the three components of velocity variance ($u'^2/2$, $v'^2/2$ and $w'^2/2$) that make up the *TKE* from all 16 sonic anemometers during the pre-burn, burn and post-burn periods. The box represents the $25^{th}$ and $75^{th}$ percentile of the data, with data inside the whiskers representing *99.3%* of the data. The orange line in the boxes is the median value, the green triangle is the mean, and the blue shading is the density of values of the data.



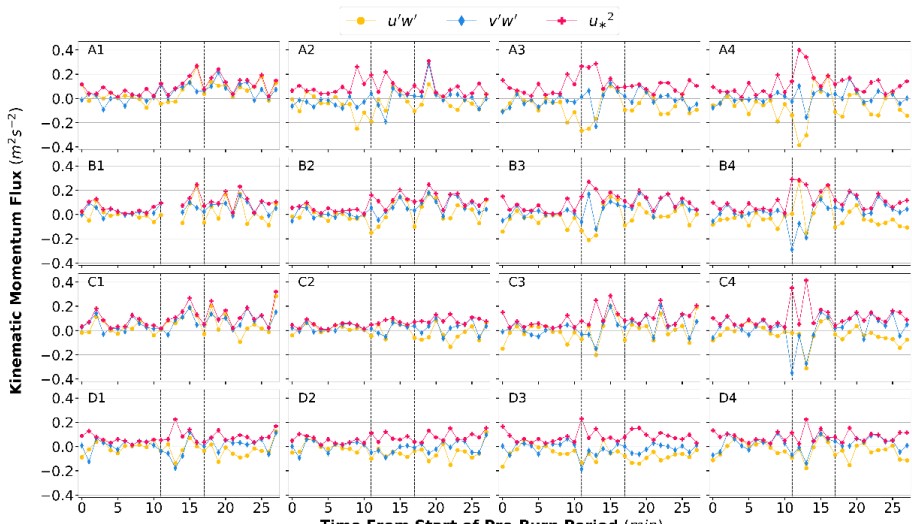

Figure 10. Time series of 1-minute averaged friction velocity squared ($u_*^2$, pink pluses) and its two components, the streamwise kinematic momentum flux, $\overline{u'w'}$ (yellow circle) and the cross-stream kinematic momentum flux, $\overline{v'w'}$ (blue diamonds), for each of the 16 sonic anemometers. The vertical dashed black lines indicate the burn period determined by the first and last occurrence of $T' \geq 8\sigma$. Time is the minutes since the start of the pre-burn period.



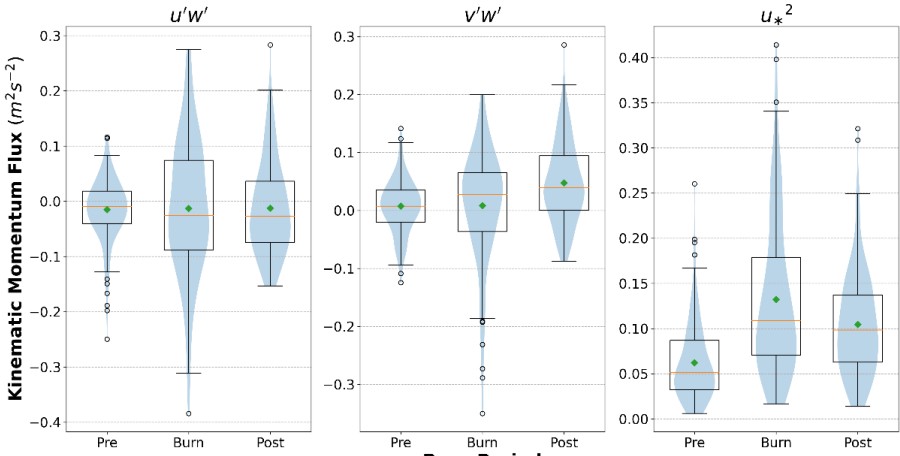

Figure 11. Distributions of friction velocity squared ($u_*{}^2$) and its two components ($\overline{u'w'}$ and $\overline{v'w'}$) from all 16 sonic anemometers during the pre-burn, burn, and post-burn periods. The box represents the $25^{th}$ and $75^{th}$ percentile of the data, with data inside the whiskers representing *99.3%* of the data. The orange line in the boxes is the median value, the green triangle is the mean, and the blue shading is the density of values of the data.



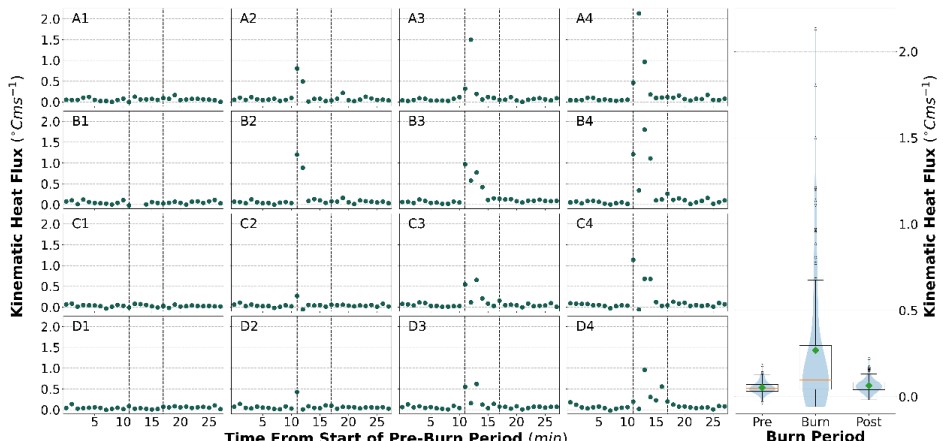

Figure 12. Time series of 1-minute averaged heat flux for each of the 16 sonic anemometers (left) and the distribution of heat fluxes from all 16 sonic anemometers during the pre-burn, burn, and post-burn periods (right). The box represents the $25^{th}$ and $75^{th}$ percentile of the data, with data inside the whiskers representing $99.3\%$ of the data. The orange line in the boxes is the median value, the green triangle is the mean, and the blue shading is the density of values of the data.



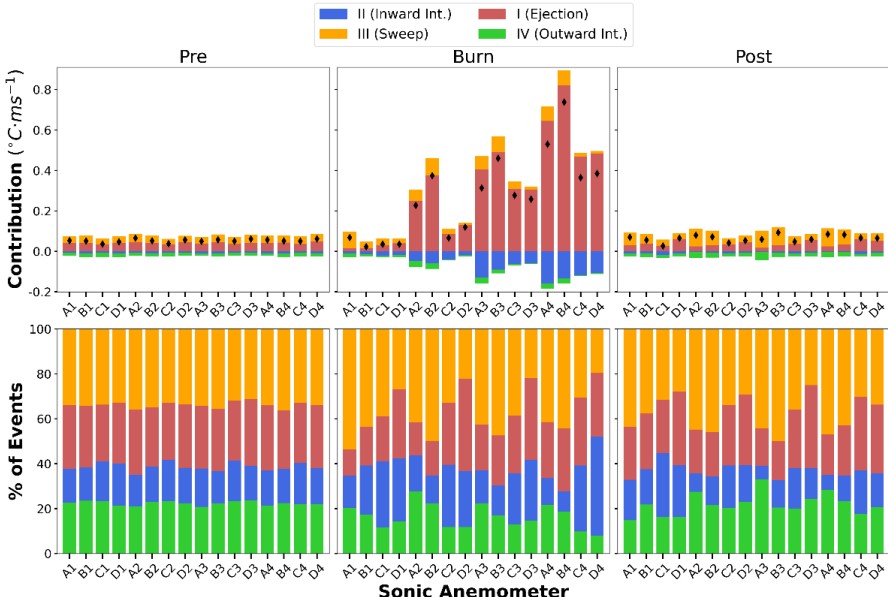

Figure 13. Quadrant analysis of the instantaneous vertical kinematic turbulent heat fluxes showing the contributions to the total flux from (top row), and the percent of (bottom row) the four types of events: outward interaction (green), ejection (red), inward interaction (blue), and sweep (orange) for each of the 16 sonic anemometers during the pre-burn, burn, and post-burn periods. The black diamonds in the top row indicate the total heat flux values. The sonic anemometers are arranged from west to east roughly following the fire spread across the burn plot.



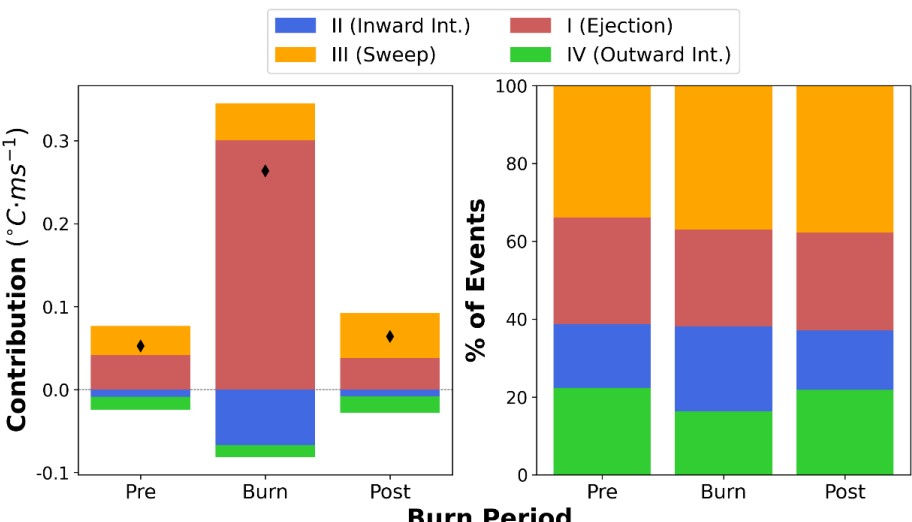

Figure 14. Quadrant analysis of the instantaneous vertical kinematic turbulent heat fluxes showing the contributions to the total flux from (top row), and the percent of (bottom row) the four types of events: outward interaction (green), ejection (red), inward interaction (blue), and sweep (orange) for all 16 sonic anemometers during the pre-burn, burn, and post-burn periods. The black diamonds in the top row indicate the total heat flux values. The sonic anemometers are arranged from west to east roughly following the fire spread across the burn plot.



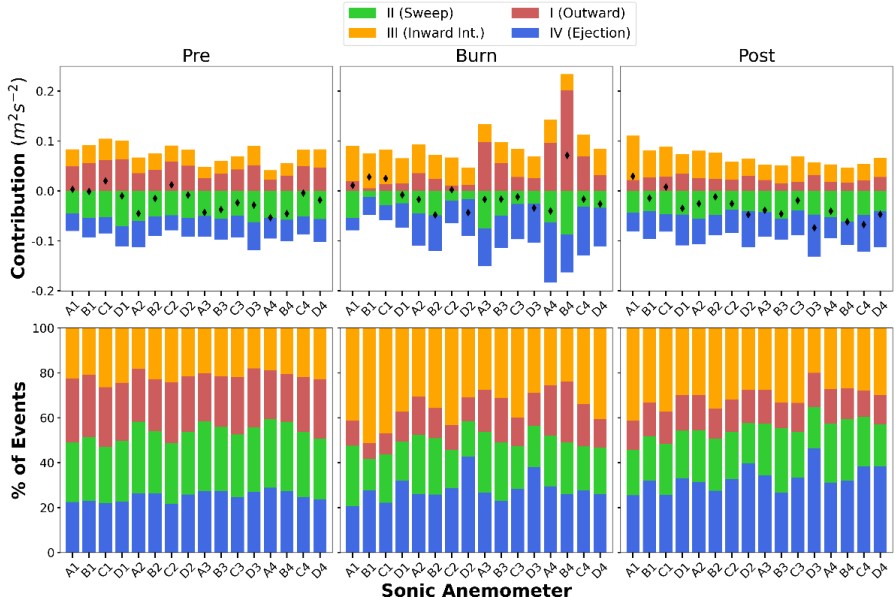

Figure 15. Quadrant analysis of the instantaneous vertical kinematic turbulent fluxes of horizontal momentum showing the contributions to the total flux from (top row), and the percent of (bottom row) the four types of events: outward interaction (red), sweep (green), inward interaction (orange), and ejection (blue) for each of the 16 sonic anemometers during the pre-burn, burn, and post-burn periods. The black diamonds in the top row indicate the total flux values. The sonic anemometers are arranged from west to east roughly following the fire spread across the burn plot.



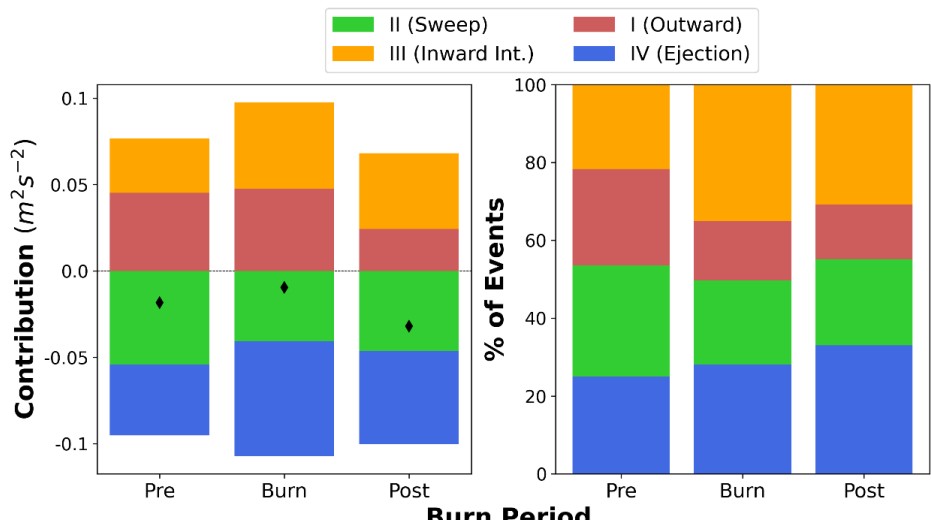

Figure 16. Quadrant analysis of the instantaneous vertical kinematic turbulent fluxes of horizontal momentum showing the contributions to the total flux from (top row), and the percent of (bottom row) the four types of events: outward interaction (red), sweep (green), inward interaction (orange), and ejection (blue) for all 16 sonic anemometers during the pre-burn, burn, and post-burn periods. The black diamonds in the top row indicate the total flux values. The sonic anemometers are arranged from west to east roughly following the fire spread across the burn plot.