# Peer review of "Atmospheric turbulence observed during a fuel-bed-scale low intensity surface fire"

_EGUsphere, 2022_

## Author Response (AR1)

We sincerely appreciate the two reviewers for their thoughtful and insightful comments on our manuscript. Their constructive feedback has played a crucial role in refining and strengthening the quality of our work. We highly appreciate the time and effort they dedicated to providing valuable suggestions, which have undoubtedly enhanced the quality of our article. Our gratitude to the reviewers has been duly acknowledged in the Acknowledgments section.

Below, we present our response (in regular font) to each of the comments (in italics).

**Reviewer 2**

*The paper describes the experimental set up and the experiment finalised at the detailed study of fire induced turbulence and subsequent data analysis.*

*The paper is very well written and the scientific contentment and relevance are high. I have no objections to the publication to the manuscript provided that some minor correction and additional comment are considered as described below.*

Response: Thank you very much for your thoughtful and positive review of our manuscript. We appreciate your kind words regarding the clarity of the writing and the perceived scientific content and relevance.

We are grateful for your constructive feedback and suggestions for improvement. We have carefully addressed the minor corrections and additional comments you have provided and we believe these revisions have substantially enhanced the overall quality of the manuscript.

*1- In my view the Introduction should be reduced or reorganized. In fact the long list of described past research activities that extends for over 5 A4 pages is less informative if one does not know what is the scope of the paper. So either the description is presented in a more concise way or the introduction starts with a brief presentation of the topic central to the paper and then the differences from the literature are presented.*

Response:  We appreciate the insightful comment regarding the Introduction. Recognizing the importance of providing readers with a clear understanding of the study's scope and objectives, we have restructured the introduction to commence with a concise presentation of the central topic and the unique contributions of our research. To address this concern, we have introduced a paragraph early in the introduction (third paragraph) that explicitly outlines the major gap in the existing literature and articulates the study's goal to fill this gap. Subsequently, we have streamlined the literature review, ensuring greater precision and relevance. We believe that these revisions significantly enhance the clarity and effectiveness of the introduction, aligning it more closely with the expectations of the readers.

*2- Whenever I read explicit research questions outlined in a introduction, which by the way is a sign of clarity in pursuing specific goals, I expect direct answer to the specific questions in the summary section. This gives normally a good sense of circularity to the manuscript and consistency in the approach*

*adopted. Listing explicitly research question is always a good way to confine and clarify the research goals – providing answers to those very same questions is a good way to complete the task.*

Response: This comment highlights an important aspect of research presentation, and we appreciate the guidance. To enhance the manuscript's coherence and consistency, we have revised the summary to explicitly connect with the three research questions outlined in the introduction. We believe this adjustment adds clarity and strengthens the overall structure of the article.

3- *In the Summary a reference is made to the inadequacy of atmospheric-fire describing models to account for fire induced turbulence whenever the grid size is larger than 1-2 m2. Since such horizontal grind resolutions are far from being adoptable especially in operational application, I wonder what is the opinion of the authors with respect to devising parametrizations. Furthermore, how much experiments such that described, can be instrumental to devise parametrizations of the process that though crude, would constitute and improvement of the current deficiency of the abovementioned models.*

Response:

We appreciate the opportunity to provide further clarification on the implications of our findings.

Our study, conducted with a sonic anemometer array on a 10 m x 10 m burn plot, revealed significant spatial variations in fire-induced perturbations and turbulent properties. This observation suggests that models employing grid spacing greater than 10 m may fail to adequately represent the spatial variability inherent in fire behavior and smoke dispersion. We acknowledge that 1-2 m horizontal resolutions are impractical for operational applications and recognize the importance of addressing this issue through appropriate parameterizations.

Fire behavior and smoke dispersion models operate across a range of spatial scales, from fine-scale to coarse-scale. Fine-scale models, with grid spacing spanning from less than a meter to several meters, are adept at capturing detailed fire spread within specific landscapes. Meanwhile, large-scale or global models, with resolutions of several to tens of kilometers, excel in simulating large-scale fire dynamics and climate interactions. However, the intermediate operational models, falling within the 'gray zone' with resolutions ranging from tens to hundreds of meters, present a unique challenge.

The 'gray zone' refers to a range of scales of model resolution where turbulence is partially resolved and existing parameterizations designed to resolve the entire scale of turbulent motions are inadequate. Advancements in computing technology have brought this zone to the forefront of operational model simulations. Developing turbulence closure schemes for this scale is an active area of research. Large-eddy simulation (LES) models, validated using laboratory data and limited field observations, are instrumental in this endeavor. The experimental data, described in our study, capturing fire-induced turbulence, can play a crucial role when combined with LES models.

Furthermore, our recent study (Kiefer et al. 2022) specifically addressed the representation of vertical heat exchange between fire and atmosphere in mesoscale models. Simulating a management-scale prescribed fire burn with an 11-hectare burn block and 30-m horizontal grid spacing, the study indicated that vertical heat exchange is largely driven by turbulent eddies unresolved by the model grid. This

underscores the need for improved parameterizations to capture turbulent processes within the gray zone.

In summary, the experiments detailed in our study offer valuable insights that can contribute to the development of parameterizations, especially in the 'gray zone' towards which the operational model simulations are rapidly approaching.

We have added several sentences to elaborate on the implication of our results for developing turbulence parameterizations in the operational models.

Reference:

Kiefer, M.T., W.E. Heilman, S. Zhong, J.J. Charney, X. Bian, N.S. Skowronski, K.L. Clark, M.R. Gallagher, J.L. Hom, and M. Patterson, 2022: Representing low-intensity fire sensible heat output in a mesoscale atmospheric model with a canopy submodel: A case study with ARPS-CANOPY (version 5.2.12). Geosci. Model Dev., 15, 1713-1734, https://doi.org/10.5194/gmd-15-1713-2022.

**Reviewer 3:**

*This paper shows how a team can produce good work. I have three points that I would like the authors to address.*

Response: Thank you for taking the time to review our manuscript. We appreciate your positive feedback on the overall demonstration of our team's ability to produce good work.

*1. Line fires have some characteristic oscillation. Was this not detected by the sonic anemometers or the temperature measurements?*

Response: Thank you for your insightful comment regarding the characteristic oscillation of line fires. Indeed, line fires exhibit oscillatory patterns. In our current manuscript, we made a deliberate choice not to delve into the analyses of potential periodicity in sonic anemometer data and temperature measurements. The inclusion of such analyses would significantly expand the scope of the paper, which already spans 40 pages of text with over 11,000 words, excluding the reference list, and includes 16 multiple-panel figures. We aimed to strike a balance between depth of analysis and the need for a manageable and accessible document.

However, your feedback has prompted us to consider a dedicated exploration of this specific aspect in a future manuscript, allowing for a more comprehensive examination of the oscillatory behavior associated with line fires. It is worth noting that our group has previously examined some periodicity patterns in momentum fluxes during a management-scale burn experiment (Heilman et al., 2021).

We have incorporated several sentences in the future work paragraph within the Summary section, highlighting our commitment to investigating the oscillatory behavior associated with line fires.

Reference:

Heilman WE, Clark KL, Bian X, Charney JJ, Zhong S, Skowronski NS, Gallagher MR, Patterson M (2021) Turbulent momentum flux behavior above a fire front in an open-canopied forest. Atmosphere 12(8), 956.

*2. Is it possible to analyse the results for the presence of vorticity and vortical structures.*

Response: We appreciate your brining up the analysis of vorticity and vortical structures associated with line fires. Recognizing their significance in understanding fire behavior, we want to acknowledge the inherent challenges in estimating these parameters from field observations.

The requirement for a meticulously designed instrument array capable of capturing horizontal variations in velocity poses a significant challenge, particularly in the context of most management-scale fire experiments where instrument density may be insufficient. However, the 4x4 sonic anemometer array utilized in the 10m x 10m burn plot presents a unique opportunity to estimate vertical vorticity associated with line fires.

We have initiated the analysis of vertical vorticity and factors contributing to its generation using the data from a series of these 10 m x 10 m burns. Our findings and insights from this investigation will be

presented in a separate manuscript, as the current focus of our work on turbulence is already quite extensive, with length constraints of the manuscript in mind.

However, it is important to note that estimating horizontal vorticity is not feasible due to the sonic anemometer array's velocity measurement on a single vertical level (2.5 m), which does not capture the necessary vertical variations of velocity for horizontal vorticity calculation. Future experiments will require deploying a densely spaced sonic anemometer similar to the current one but at multiple vertical levels to comprehensively evaluate vorticity associated with fires.

We have dedicated a paragraph in the "Future Work" section of the Summary to address the challenges associated with vorticity estimation and highlight the unique opportunity presented by this dataset for gaining insights into fire-induced vertical vorticity.

*3. I think that the major disturbance from the fire is caused on the environment near the ground. Why no attempt was made to study this effect? Maybe it has been done elsewhere.*

Response: We appreciate your comment and seek clarification on whether you are suggesting that the environment near the ground causes the major disturbance from the fire or if you are indicating that the major disturbance from the fire exerts its largest impact near the ground.

In our analyses presented in this manuscript, our primary focus was on observations from the array of 3D sonic anemometers mounted at 2.5 m above the ground. This elevation was chosen strategically to avoid potential damages from the fire while still capturing critical data. As outlined in the experiment and instrument section, we also collected 10-Hz temperature data using fine-wire thermocouples (Omega SSRTC-GG-K-36, Omega Engineering, Inc., Stamford, CT, USA) at various heights (0, 5, 10, 20, 30, 50, 100 cm) below the two inner trusses (B and C). These data enable us to assess temperature changes near the ground as the fire spreads across the burn plot. The vertical temperature profiles captured by the thermocouples, spanning from the combustion zone to 1 m above, will be especially valuable for estimating the vertical temperature gradient—a crucial variable in the parameterization of turbulence heat flux.

Although we opted not to include thermocouple data analyses in the current manuscript, emphasizing our focus on fire-induced turbulent stress and fluxes, the results from the analyses of thermocouple data, combined with infrared data highlighting fire spread and near-ground temperature changes, will be presented in subsequent studies. We have included a sentence in the Summary section to convey this information.